# FEDERATED DYNAMICAL LOW-RANK TRAINING WITH GLOBAL LOSS CONVERGENCE GUARANTEES

## ABSTRACT

We propose a federated dynamical low-rank training (FeDLRT) scheme to reduce client compute and communication costs - two significant performance bottlenecks in horizontal federated learning. Our method builds upon dynamical low-rank splitting schemes for manifold-constrained optimization to create a global low-rank basis of network weights, which enables client training on a small coefficient matrix. This global low-rank basis that allows us to incorporate a variance correction scheme and prove global loss descent and convergence to a stationary point. FeDLRT features dynamic augmentation and truncation of the low-rank bases to optimize computing and communication resource utilization. Notably FeDLRT only trains a small coefficient matrix per client. We demonstrate the efficiency of FeDLRT in an array of computer vision benchmarks with both i.i.d. and non-i.i.d. data distributions and show a reduction of client compute and communication costs by up to an order of magnitude with minimal impacts on global accuracy. FeDLRT performs as well as classical methods such as FedAvg and FedLin, with a fraction of the memory and compute requirements.

## 1 INTRODUCTION

Federated learning (FL) Li et al. (2020); Shamir et al. (2014); McMahan et al. (2016) builds a global model on a central *server* from data distributed on multiple devices, i.e., *clients*, by iteratively aggregating local models trained with the computation resource on the clients. In horizontal FL, where all clients share identical model architecture and data features, computation is often limited by (i) the communication bandwidth between clients and the server and (ii) the restricted compute and memory resources at each client. The former could be addressed by deploying various compression techniques, such as sparse randomized sketching Haddadpour et al. (2020b), subsampling Konečný et al. (2017), or by allowing for partial McMahan et al. (2016); Nishio & Yonetani (2019) or asynchronous Sprague et al. (2018); Chen et al. (2020b) communications. The latter could be addressed by sparse training Qiu et al. (2022); Yang et al. (2020) and transfer learning Chen et al. (2020a).

Since FedAvg McMahan et al. (2016), low-rank, sparsity and matrix sketching-based methods have been proposed to increase communication and compute efficiency for FL in Qiao et al. (2021); Yi et al. (2024); Liu et al. (2023); Yao et al. (2022); Xue & Lau (2023); Hyeon-Woo et al. (2022); Konečný et al. (2017); Reisizadeh et al. (2020b). These methods can be categorized into 1) methods that perform full-rank training on the clients and reduce communication cost by communicating only a) low-rank factors Qiao et al. (2021); Vogels et al. (2020); Xue & Lau (2023) or b) sketched matrices Rabbani et al. (2023); Rothchild et al. (2020); Ivkin et al. (2019); Condat et al. (2023) and 2) methods that reduce both communication and client compute costs by training on a) low-rank factors Liu et al. (2023); Yi et al. (2024); Yao et al. (2022); Hyeon-Woo et al. (2022); Konečný et al. (2017); Coquelin et al. (2024) or b) sparsity patterns Horváth et al. (2021) on the clients. The methods in the first class compress only the communication and do not reduce the compute and memory cost, while the ones in the second class reduces the client compute and memory cost but often require reconstructing the full weight matrix on the server.

Further, application of multiple optimization steps (local iterations) on clients often leads to the client drift phenomenon, where convergence to local minimizers stalls the global convergence. Several methods Shamir et al. (2014); Li et al. (2020); Pathak & Wainwright; Karimireddy et al. (2020); Wang et al. (2020); Mitra et al. (2021b) have been proposed to mitigate this issue for non-compressed

Table 1: Comparison of the computational footprint of FeDLRT with FedAvg McMahan et al. (2016), FedLin Mitra et al. (2021a), FeDLR Qiao et al. (2021), Riemannian FL Xue & Lau (2023), FjORD Horváth et al. (2021), FetchSGD Rothchild et al. (2020), FedHM Yao et al. (2022) and FedPara Hyeon-Woo et al. (2022). We denote the number of local iterations by $s_*$, the local batch size as $b$, the matrix dimension by $n \times n$ and the matrix rank by $r$. The sparsity rate of FjORD and FetchSGD is denoted by $\delta$. We mark the low-rank method with lowest compute, memory and communication cost blue assuming fixed $r \ll n$. The FeDLRT variants are the only low-rank schemes with linearly scaling (in $n$) memory, compute, and communication costs with automatic compression and the ability to handle client drift .

| Method | Client compute | Client memory | Server compute | Server memory | Comm. cost | Comm. rounds | Handles client drift | Rank adaptive |
|---|---|---|---|---|---|---|---|---|
| FedAVG | $\mathcal{O}(s_*bn^2)$ | $\mathcal{O}(2n^2)$ | $\mathcal{O}(n^2)$ | $\mathcal{O}(2n^2)$ | $\mathcal{O}(2n^2)$ | 1 | ✗ | ✗ |
| FedLin | $\mathcal{O}(s_*bn^2)$ | $\mathcal{O}(2n^2)$ | $\mathcal{O}(n^2)$ | $\mathcal{O}(2n^2)$ | $\mathcal{O}(4n^2)$ | 2 | ✓ | ✗ |
| FeDLR | $\mathcal{O}(s_*bn^2 + n^3)$ | $\mathcal{O}(2n^2)$ | $\mathcal{O}(n^3 + n^2)$ | $\mathcal{O}(n^2)$ | $\mathcal{O}(4nr)$ | 1 | ✗ | ✓ |
| Riemannian FL | $\mathcal{O}(s_*bn^2r)$ | $\mathcal{O}(2n^2)$ | $\mathcal{O}(n^2r)$ | $\mathcal{O}(4nr)$ | $\mathcal{O}(4nr)$ | 1 | ✗ | ✓ |
| FedHM | $\mathcal{O}(s_*b2nr)$ | $\mathcal{O}(2nr)$ | $\mathcal{O}(n^2 + n^3)$ | $\mathcal{O}(n^2)$ | $\mathcal{O}(4nr)$ | 1 | ✗ | ✓ |
| FedPara | $\mathcal{O}(s_*b4nr)$ | $\mathcal{O}(4nr)$ | $\mathcal{O}(n^2 + n^3)$ | $\mathcal{O}(2n^2)$ | $\mathcal{O}(8nr)$ | 1 | ✗ | ✗ |
| FjORD | $\mathcal{O}(s_*b\delta)$ | $\mathcal{O}(\delta)$ | $\mathcal{O}(n^2)$ | $\mathcal{O}(n^2)$ | $\mathcal{O}(4\delta)$ | 1 | ✗ | adapts $\delta$ |
| FetchSGD ($s_* = 1$) | $\mathcal{O}(s_*bn^2)$ | $\mathcal{O}(\delta)$ | $\mathcal{O}(n^2)$ | $\mathcal{O}(n^2)$ | $\mathcal{O}(4\delta)$ | 1 | for $s_* = 1$ | adapts $\delta$ |
| FeDLRT w/o var/cor | $\mathcal{O}(s_*b4nr)$ | $\mathcal{O}(4nr)$ | $\mathcal{O}(4nr^2)$ | $\mathcal{O}(2nr)$ | $\mathcal{O}(6nr + 6r^2)$ | 2 | ✗ | ✓ |
| FeDLRT simpl. var/cor | $\mathcal{O}(s_*b4nr)$ | $\mathcal{O}(4nr)$ | $\mathcal{O}(4nr^2)$ | $\mathcal{O}(2nr)$ | $\mathcal{O}(6nr + 8r^2)$ | 2 | ✓ | ✓ |
| FeDLRT full var/cor | $\mathcal{O}(s_*b4nr)$ | $\mathcal{O}(4nr)$ | $\mathcal{O}(4nr^2)$ | $\mathcal{O}(2nr)$ | $\mathcal{O}(6nr + 10r^2)$ | 3 | ✓ | ✓ |

models, often by introducing correction terms to the client gradient. However, applying these client drift mitigation techniques to methods in the second class is nontrivial, since the correction term is often not compatible with the compressed (low rank or sparse) representations in local training.

**Contribution:** This work focuses on the horizontal FL setting and addresses the challenges of communication bandwidth and client compute resources simultaneously by leveraging low-rank approximations of weight matrices that follow the dynamics of the gradient flow. The proposed method features 1) **Efficient communication** — only transmitting low-rank factors; 2) **Low client compute and memory footprint** — clients optimizing only a small coefficient matrix and all floating point operations of FeDLRT scale linearly in the matrix dimension $n$; 3) **Automatic server-side compression** — minimizing memory and communication requirements during training via server-side dynamical rank adjustment; 4) **Global loss convergence guarantees** — converging to a stationary point by incorporating a variance correction scheme Mitra et al. (2021a). Each of these features is demonstrated on benchmark problems. To the best of the authors' knowledge, this is the first low-rank method possessing all these features.

## 2 BACKGROUND AND PROBLEM STATEMENT

**Federated optimization** typically considers *distributed* setups and with *limited communication* and *limited client compute and memory* resources McMahan et al. (2016). In this work, we consider a general federated optimization problem, i.e.,

$$\min_w \mathcal{L}(w) := \frac{1}{C} \sum_{c=1}^{C} \mathcal{L}_c(w), \tag{1}$$

where $w$ is a trainable weight, $\mathcal{L}$ is the global loss function associated to a global dataset $X$, and $\mathcal{L}_c$ is the local loss function of client $c$ with local dataset $X_c$ in a federated setup with $C$ clients. For notational simplicity, we consider that $X = \cup_{c=1}^{C} X_c$ and each $X_c$ is of the same size. Therefore, $\mathcal{L}$ is an average of $\mathcal{L}_c$ with uniform weights.

The extension to handle a (non-uniform) weighted average case is straightforward. As the first baseline for federated optimization, we consider FedAvg McMahan et al. (2016), see Algorithm 3. Here, each client optimizes its local loss function $\mathcal{L}_c$ for $s_*$ local iterations using gradient descent,

$$w_c^{s+1} = w_c^s - \lambda \nabla_w \mathcal{L}(w_c^s), \tag{2}$$

with learning rate $\lambda$, for $s = 0, \dots, s_* - 1$. The initial value for the local iteration is the last global weight, i.e., $w_c^0 = w^t$. After local iterations, the weights are communicated to and aggregated at the server to update the global weight following

$$w^{t+1} = \frac{1}{C} \sum_{c=1}^{C} w_c^{s_*}. \qquad (3)$$

**Client-drift effect** is a common challenge in FL, where the iterative client updates (2) of FedAvg converge to local minima and jeopardize global training performance since the average of the local minimizers may be far away from the global minimizer. These effects are particularly pronounced for a large number of local iterations $s_*$, or high discrepancies between local loss functions $\mathcal{L}_c$, as illustrated by Figure 1. Multiple methods Shamir et al. (2014); Li et al. (2020); Pathak & Wainwright; Karimireddy et al. (2020); Wang et al. (2020) have been proposed to mitigate this issue. However, these methods often exhibit a *speed-accuracy conflict*, where learning rates need to be heavily reduced; thus, convergence is slow.

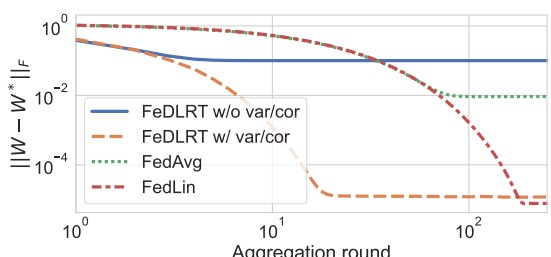

Figure 1: Federated, heterogeneous least squares regression problem, see Section 4.1, for $C = 4$ clients, $s_* = 100$ iterations, learning rate $\lambda = 1\mathrm{e} - 3$ and $C$ rank-1 local target functions. FL methods without variance correction plateau quickly, whereas FedLin and FeDLRT with variance correction converge to $1\mathrm{e} - 5$. FeDLRT converges faster than FedLin and has lower communication costs.

**Variance correction**[1] introduced in the FedLin method Mitra et al. (2021a) constructs a variance correction term $V_c = \nabla_w \mathcal{L}_c(w^t) - \frac{1}{C} \sum_{c=1}^{C} \nabla_w \mathcal{L}_c(w^t)$ and modifies the client update iteration to

$$w_c^{s+1} = w_c^s - \lambda \left( \nabla_w \mathcal{L}(w_c^s) - V_c \right), \qquad s = 0, \ldots, s_* - 1. \qquad (4)$$

This technique leads to global convergence to the minimizer of (1) with constant learning rates for convex $\mathcal{L}$ and else to convergence to a stationary point, at the cost of an additional communication round for computing the variance correction. Similar methods, sometimes dubbed "error feedback" are proposed in Liang et al. (2019); Ivkin et al. (2020)

**Federated neural network training** considers problem (1) with the trainable weight $w$ being the set of weight matrices $\{W_i\}_i^L$ of an $L$ layer neural network. In each iteration, the weight updates in (2) and (4) are applied to all layers simultaneously. Therefore, w.l.o.g., we express the local loss function as $\mathcal{L}_c(W)$, where $W \in \mathbb{R}^{n \times n}$ denotes the weight matrix of an arbitrary layer.

**Low-rank neural network training:** An array of recent work has provided theoretical and experimental evidence that layer weights of over-parameterized networks tend to be low rank Arora et al. (2019); Bah et al. (2022); Galanti et al. (2022); Martin & Mahoney (2018) and that removing small singular values may even lead to increased model performance while dramatically reducing model size Sharma et al. (2024); Schotthöfer et al. (2022) in non-federated scenarios. This beneficial feature has spawned a rich landscape of methods to compress neural networks to a low-rank factorization after training with subsequent fine-tuning Sainath et al. (2013); Denton et al. (2014); Tjandra et al. (2017); Lebedev et al. (2015), train the factorized network with fixed rank Jaderberg et al. (2014); Wang et al. (2021); Khodak et al. (2021), dynamically adjust the rank during training Schotthöfer et al. (2022); Zangrando et al. (2023), or use low-rank adapters for fine-tuning foundation models Hu et al. (2021); Dettmers et al. (2023); Zhao et al. (2024); Schotthöfer et al. (2024).

**Dynamical Low-rank Approximation of the gradient flow of neural network training**. The core contribution of this paper builds on the dynamical low-rank approximation (DLRA) method, which was initially proposed for solving matrix equations Koch & Lubich (2007) and recently extended to neural network training Schotthöfer et al. (2022); Zangrando et al. (2023); Hnatiuk et al. (2024); Schotthöfer et al. (2024). Let $\dot{W}(t) = -\nabla_W \mathcal{L}(W(t))$ denote the gradient flow for minimizing $\mathcal{L}$.

The DLRA method restricts the trajectory of $W$ to $\mathcal{M}_r$, the manifold of $n \times n$, rank-$r$ matrices, by projecting $\dot{W}$ onto a local tangent plane of $\mathcal{M}_r$ via an orthogonal projection. This guarantees a low-rank solution when following the projected dynamics from a low-rank initial guess. Let the low-rank matrix take the form $W_r = USV^\top \in \mathcal{M}_r$ with $U, V \in \mathbb{R}^{n \times r}$ the orthonormal bases of

---

[1]Variance correction is commonly referred to as "variance reduction" Konečný et al. (2016); Mitra et al. (2021a).

$\mathcal{M}_r$ and $S \in \mathbb{R}^{r \times r}$ the coefficient matrix. The dynamics for each low-rank factor in DRLA are then derived in (Koch & Lubich, 2007, Proposition 2.1) as

$$
\begin{aligned}
\dot{S}(t) &= -U^\top(t) \nabla_W \mathcal{L}(U(t)S(t)V(t)^\top)V(t), \\
\dot{U}(t) &= -\left(I - P_{U(t)}\right) \nabla_W \mathcal{L}(U(t)S(t)V(t)^\top)V(t)S(t)^{-1}, \\
\dot{V}(t) &= -\left(I - P_{V(t)}\right) \nabla_W \mathcal{L}(U(t)S(t)V(t)^\top)U(t)S(t)^{-\top},
\end{aligned}
\tag{5}
$$

where $P_U = UU^\top$ and $P_V = VV^\top$ are the projections onto the column spaces of $U$ and $V$, respectively.

In Schotthöfer et al. (2022), the authors develop an memory and compute efficient scheme to solve (5) numerically for (non-federated) neural network training. The idea is to split the system into a basis update step for $U$ and $V$ and a coefficient update step for $S$, allowing rank adaptation via a basis augmentation before the coefficient update step and a basis truncation after the coefficient update, which enables dynamic compression of the neural network during training.

**Direct extension of DLRA to FL is not straightforward**: In a FL setup, applying the above scheme to the local training problem on each client $c$ leads to low-rank weights $W_c = U_c S_c V_c^\top$ with different bases $U_c$ and $V_c$ and potentially different ranks for each client. While these factors can still be efficiently communicated, aggregating these low-rank weights on the server requires reconstructing the full weight matrix $W^* = \frac{1}{C} \sum_{c=1}^{C} U_c S_c V_c^\top$. In this process, the low rank structure is lost and needs to be costly recovered by a full $n \times n$ SVD on the server. Low-rank schemes using this type of aggregation step and direct gradient descent on low-rank factors are presented in, e.g. Yao et al. (2022), Qiao et al. (2021), Hyeon-Woo et al. (2022).

Furthermore, the inconsistency of the low-rank bases on each client complicates the implementation of standard client drift mitigation methods, e.g., the variance correction method in FedLin Mitra et al. (2021b), since these methods require averaged gradient information of all clients, which is nontrivial to compute without a global low-rank basis shared between clients.

## 3 FeDLRT: Federated dynamical low-rank training with variance correction

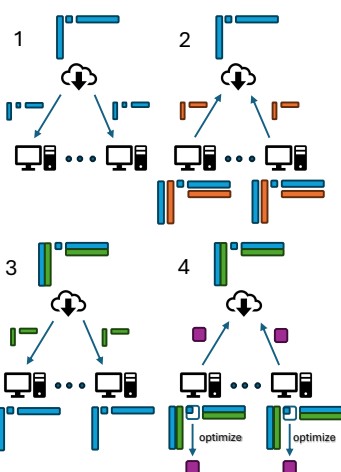

In this section, we present the core contribution of this paper, *federated dynamical low-rank training* (FeDLRT). FeDLRT creates a global low-rank manifold on which all clients of the FL setup share the same basis $U, V$, referred to as the global basis. This core idea enables FeDLRT to reduce the compute, communication, and memory costs simultaneously while incorporating client drift mitigation techniques to guarantee convergence.

The procedure of FeDLRT is illustrated in Figure 2 and detailed in Algorithm 1. As shown in Figure 2, FeDLRT first broadcasts an initial low-rank factorization of a weight matrix $W_r = USV^\top$ to the clients (panel 1), and the basis gradients[2] $U, V$ are aggregated on the server (panel 2). Next, the basis is augmented on the server (panel 3) and broadcast. On the clients, only the augmented coefficient matrix $S$ is updated repeatedly (panel 4) before aggregation to the server. After aggregation of the local augmented coefficient matrices, redundant bases are eliminated to optimize the accuracy-to-compression ratio of the model on the server, which gives the low-rank factorization of the global weight for the broadcasting step in panel 1 for the next aggregation round.

Figure 2: Communication of FeDLRT without variance correction. 1) Broadcast global basis $U, V$ (blue). 2) Aggregate basis gradients $G_{c,U}, G_{c,V}$ (orange). 3) Broadcast global augmented basis $\bar{U}, \bar{V}$ (green). 4) Aggregate client coefficient update $\widetilde{S}_c^{s*}$ (purple).

The strategy yields the following benefits compared to "full-rank" FL schemes, such as FedLin Mitra et al. (2021a):

**Low client compute cost:** Server-based basis augmentation and

---

[2] and later on the coefficient gradients for variance correction

compression enables an automatic compression without a-priori knowledge of the layer rank $r$ and at no cost for the resource-constrained clients. The clients only evaluate gradients of low-rank factors and optimize the small matrix $S \in \mathbb{R}^{r \times r}$.

**Efficient communication:** Similar to FedLin, FeDLRT requires *in practice* two communication rounds – one for aggregating and distributing global gradients for basis augmentation and variance correction and one for aggregating locally updated coefficients. However, communication cost for each round is significantly reduced since only low-rank factors are communicated. We refer to Section 3.3 on communication and compute cost.

In comparison to low-rank schemes with local compression Yao et al. (2022); Yi et al. (2024); Qiao et al. (2021); Hyeon-Woo et al. (2022) the scheme has the following benefits:
**Global manifold basis:** Splitting the low-rank update and sharing bases among clients provides a globally consistent manifold basis. The global basis unlocks the favorable features of dynamical low-rank training Schotthöfer et al. (2022) for federated learning: 1) Convergence behavior that mimics that of FedLin, 2) compute and memory efficiency by never computing or assembling full weight matrices, 3) automatic compression through rank adaptivity instead of manual tuning of the rank $r$.

## 3.1 DESCRIPTION OF ALGORITHM 1 - FeDLRT

In this section, we elaborate on the details in Algorithm 1[3]. The orthonormal factors $U^t, V^t$ and the coefficient matrix $S^t$ are initialized with rank $r$ and then broadcast to the clients. Note that FeDLRT ensures that, for all $t > 1$, $U^t$ and $V^t$ are orthonormal, and $S^t$ is diagonal and full rank.

**Basis augmentation** of the bases $U^t$ and $V^t$ is performed using concatenation with the corresponding global basis gradients $G_U = \frac{1}{C} \sum_{c=1}^{C} \nabla_U \mathcal{L}_c(U^t S^t V^{t,\top})$ and $G_V = \frac{1}{C} \sum_{c=1}^{C} \nabla_V \mathcal{L}_c(U^t S^t V^{t,\top})$, obtained by aggregating the local basis gradients. $G_U$ and $G_V$ encapsulate the gradient flow dynamics (5) projected onto the original bases, thus yielding an intuitive choice for basis augmentation. Further, this choice is consistent with the basis update step of the augmented BUG splitting scheme, see Appendix F, which ensures the robustness of the client optimizer. Subsequent orthonormalization, e.g., by a QR decomposition, yields the augmented basis, i.e.,

$$[U^t \mid \bar{U}]R = \texttt{qr}([U^t \mid G_U]) \in \mathbb{R}^{n \times 2r}, \quad \text{and} \quad [V^t \mid \bar{V}]R = \texttt{qr}([V^t \mid G_V]) \in \mathbb{R}^{n \times 2r}. \quad (6)$$

We denote the augmented bases by $\widetilde{U} = [U^t \mid \bar{U}]$ and $\widetilde{V} = [V^t \mid \bar{V}]$. The orthonormalization is performed on the server, providing compute cost reduction for the client.

**Basis broadcasting** of $\widetilde{U}$ and $\widetilde{V}$ only requires to broadcast the new bases $\bar{U}$ and $\bar{V}$, since $U^t$ and $V^t$ are readily available on the clients. Formally, the coefficients $S^t$ are projected onto the augmented basis, i.e., $\widetilde{S} = \widetilde{U}^\top U^t S^t V^{t,\top} \widetilde{V} \in \mathbb{R}^{2r \times 2r}$, before broadcasting them to the clients. Exploiting the orthonormality of the basis results in further reduction of the communication and compute cost:

**Lemma 1.** *Let $\widetilde{U} = [U^t \mid \bar{U}]$ and $\widetilde{V} = [V^t \mid \bar{V}]$, then $\widetilde{S} := \widetilde{U}^\top U^t S^t V^{t,\top} \widetilde{V} = \begin{bmatrix} S^t & 0 \\ 0 & 0 \end{bmatrix}$.*

The proof (see Appendix G) is based on the orthogonality imposed in (6). With Lemma 1, only $\bar{U}$ and $\bar{V}$ have to be broadcast, and the augmented bases and coefficients $\widetilde{U}$, $\widetilde{V}$, and $\widetilde{S}$ can be assembled on each client as needed. Furthermore, only $S \in \mathbb{R}^{r \times r}$, instead of $\widetilde{S} \in \mathbb{R}^{2r \times 2r}$, needs to be communicated.

Below, we discuss three options for the client coefficient update step.

**Client coefficient update** without variance correction is implemented similarly to FedAvg (3). On each client $c$, the augmented coefficient matrix $\widetilde{S}_c$ is trained for $s_*$ iterations[4] with learning rate $\lambda$,

$$\widetilde{S}_c^{s+1} = \widetilde{S}_c^s - \lambda \nabla_{\widetilde{S}} \mathcal{L}_c(\widetilde{U}\widetilde{S}_c^s \widetilde{V}^\top), \quad s = 0, \ldots, s_* - 1, \quad \text{with} \quad \widetilde{S}_c^{s=0} = \widetilde{S}. \quad (7)$$

**Client coefficient update with variance correction** is required in certain federated scenarios, e.g., the case considered in Figure 1. Based on FedLin Mitra et al. (2021a), we introduce a correction

---

[3]The auxiliary functions for Algorithm 1 can be found in Algorithm 2.

[4]Our analysis focuses on the case where all clients share the same number of local iterations $s_*$. The analysis can be extended to the case where $s_*$ is client dependent, following a similar strategy as in Mitra et al. (2021a).

---

**Algorithm 1:** FeDLRT (See Algorithm 2 for auxiliary function definitions)

---

**Input :** Initial orthonormal bases $U^1, V^1 \in \mathbb{R}^{n \times r}$ and full rank $S^1 \in \mathbb{R}^{r \times r}$;
Client-server setup with clients $c = 1, \ldots, C$;
`var_cor`: Boolean flag to activate variance correction;
$\tau$: singular value threshold for rank truncation.

1 **for** $t = 1, \ldots, T$ **do**
2     `broadcast`($\{U^t, V^t, S^t\}$)
3     $G_{U,c} \leftarrow \nabla_U \mathcal{L}_c(U^t S^t V^{t,\top}); G_{V,c} \leftarrow \nabla_V \mathcal{L}_c(U^t S^t V^{t,\top})$       /\* On client \*/
4     $G_U, G_V \leftarrow$ `aggregate`($\{G_{U,c}, G_{V,c}\}$)
5     $\bar{U} \leftarrow$ `basis_augmentation`($U^t, G_U$); $\bar{V} \leftarrow$ `basis_augmentation`($V^t, G_V$)
6     `broadcast`($\{\bar{U}, \bar{V}\}$)
7     $\widetilde{U} \leftarrow [U^t \,|\, \bar{U}]; \widetilde{V} \leftarrow [V^t \,|\, \bar{V}]$       /\* Basis assembly on client \*/
8     $\widetilde{S}^{s=0} \leftarrow \begin{bmatrix} S^t & 0 \\ 0 & 0 \end{bmatrix}$       /\* Coefficient matrix assembly on client \*/
9     **if** `var_cor` **then**
10       $G_{\widetilde{S},c} \leftarrow \nabla_{\widetilde{S}} \mathcal{L}_c(\widetilde{U} \widetilde{S} \widetilde{V}^\top)$       /\* Augmented gradient on client \*/
11       $G_{\widetilde{S}} \leftarrow$ `aggregate`($\{G_{\widetilde{S},c}\}$)
12       `broadcast`($\{G_{\widetilde{S}}\}$)
13       `coefficient_update_var_cor`($c, G_{\widetilde{S}} - G_{\widetilde{S},c}$)       /\* On client \*/
14     **else**
15       `coefficient_update`($c$)       /\* On client \*/
16     $\widetilde{S}^* \leftarrow$ `aggregate`($\{\widetilde{S}_c^{s*}\}$)
17     $P_{r_1}, \Sigma_{r_1}, Q_{r_1} \leftarrow$ `svd`($\widetilde{S}^*$) with threshold $\vartheta$       /\* Compression step \*/
18     $U^{t+1} \leftarrow \widetilde{U} P_{r_1}; V^{t+1} \leftarrow \widetilde{V} Q_{r_1}; S^{t+1} \leftarrow \Sigma_{r_1}$       /\* Basis and coefficient update \*/

---

step for the local coefficient update of FeDLRT. It extends the above local iteration by another communication round, where the gradient of the augmented coefficients $G_{\widetilde{S},c} = \nabla_{\widetilde{S}} \mathcal{L}_c(\widetilde{U} \widetilde{S} \widetilde{V}^\top)$ is computed, aggregated to $G_{\widetilde{S}} = \frac{1}{C} \sum_{c=1}^C G_{\widetilde{S},c}$ and subsequently broadcast. This yields a correction term $V_c = G_{\widetilde{S}} - G_{\widetilde{S},c}$ for each client $c$ and thus the client iterations read

$$\widetilde{S}_c^{s+1} = \widetilde{S}_c^s - \lambda \left( \nabla_{\widetilde{S}} \mathcal{L}_c(\widetilde{U} \widetilde{S}_c^s \widetilde{V}^\top) + V_c \right), \quad s = 0, \ldots, s_* - 1, \quad \text{with} \quad \widetilde{S}_c^{s=0} = \widetilde{S}. \quad (8)$$

The correction term results in a bound on the coefficient drift and leads to convergence guarantees for FeDLRT, as detailed in Section 3.2.

**Client coefficient update with simplified variance correction**: Empirically, we observe that a simplified variance correction, which only considers the correction term of the *non-augmented* coefficients $S^t$, is sufficient, see Figure 8. The simplified variance correction term takes the form

$$V_c = G_{\widetilde{S}} - G_{\widetilde{S},c} \approx \check{V}_c := \check{G}_{\widetilde{S}} - \check{G}_{\widetilde{S},c} = \begin{bmatrix} \nabla_S \mathcal{L}(U^t S^t V^{t,\top}) - \nabla_S \mathcal{L}_c(U^t S^t V^{t,\top}) & 0 \\ 0 & 0 \end{bmatrix}, \quad (9)$$

which makes lines 10 and 12 in Algorithm 1 redundant, since $\check{G}_{\widetilde{S}}$ can be aggregated in one step with the basis gradients $G_U, G_V$ in line 4 and broadcast with $\bar{U}, \bar{V}$ in line 6, reducing the communication rounds to two - the same as FedLin. See Algorithm 5 for details.

**Coefficient averaging** is performed after (any of the above variants of) the client iterations. The server computes the updated global coefficients by averaging the local updates, i.e., $\widetilde{S}^* = \frac{1}{C} \sum_{c=1}^C \widetilde{S}_c^{s_*}$. With the shared augmented bases $\widetilde{U}$ and $\widetilde{V}$, this is equivalent to the FedAvg aggregation

$$\widetilde{W}_r^* = \frac{1}{C} \sum_{c=1}^C \widetilde{W}_r^{s*} = \frac{1}{C} \sum_{c=1}^C \left( \widetilde{U} \widetilde{S}_c^{s*} \widetilde{V}^\top \right) = \widetilde{U} (\frac{1}{C} \sum_{c=1}^C \widetilde{S}_c^{s*}) \widetilde{V}^\top = \widetilde{U} \widetilde{S}^* \widetilde{V}^\top. \quad (10)$$

Since the basis is fixed, the rank $2r$ is preserved in the aggregation, which is in contrast to other federated low-rank schemes where the aggregated weights could be full rank and, in turn, require a full matrix SVD to determine the new rank Qiao et al. (2021); Xue & Lau (2023).

**Automatic compression via rank truncation** is necessary 1) to identify the optimal rank of the weight matrix and 2) to ensure that $S$ is full rank[5]. To this end, a truncated SVD of $\widetilde{S}^* \in \mathbb{R}^{2r \times 2r}$ is performed, i.e. $P_{r_1}, \Sigma_{r_1}, Q_{r_1}^\top = \mathrm{svd}(\widetilde{S}^*)$, where $P_{r_1}, Q_{r_1} \in \mathbb{R}^{2r \times r_1}$ and $\Sigma_{r_1} = \mathrm{diag}(\sigma_1, \ldots, \sigma_{r_1})$ contains the $r_1$ largest singular values of $\widetilde{S}^*$. The new rank $r_1$ can be chosen by a variety of criteria, e.g., a singular value threshold $\|[\sigma_{r_1}, \ldots, \sigma_{2r}]\|_2 < \vartheta$. Once a suitable rank is determined, the factorization is updated by the projection of the bases $U^{t+1} = \widetilde{U} P_{r_1} \in \mathbb{R}^{n \times r_1}$, $V^{t+1} = \widetilde{V} Q_{r_1} \in \mathbb{R}^{n \times r_1}$ and update of the coefficient $S^{t+1} = \Sigma_{r_1}$. Remarkably, Algorithm 1 is a federated low-rank learning scheme whose solution is close to a full-rank solution, see Theorem 5.

FeDLRT can readily be extended to tensor-valued, e.g., convolutional, layers by applying Algorithm 1 to each basis and the core tensor in a Tucker Tensor factorization. We refer to Appendix B for details.

## 3.2 ANALYSIS OF FeDLRT WITH VARIANCE CORRECTION

In this section, we analyze the FeDLRT algorithm under the general assumption that $\mathcal{L}_c$ and $\mathcal{L}$ are $L$-smooth with constant $L$. Theorems 2 and 3 give the convergence results for FeDLRT with full variance correction (8) in Algorithm 1. Theorem 4 and Corollary 1 provide the convergence for FeDLRT with simplified variance correction in (9), as detailed in Algorithm 5, under additional assumptions given therein. We note that the analysis does not require convexity of $\mathcal{L}_c$ or $\mathcal{L}$.

**FeDLRT convergence with full variance correction.** The variance-corrected client iteration (8) leads to the following bound the client coefficient drift.

**Theorem 1.** *Given augmented basis and coefficient matrices $\widetilde{U}$, $\widetilde{V}$, and $\widetilde{S}$. If the local learning rate $0 < \lambda \leq \frac{1}{Ls_*}$ with $s_* \geq 1$ the number of local steps, for all clients $c$,*

$$\|\widetilde{S}_c^s - \widetilde{S}_c\| \leq \exp(1) s_* \lambda \|\nabla_{\widetilde{S}} \mathcal{L}(\widetilde{U} \widetilde{S} \widetilde{V}^\top)\|, \quad \text{for} \quad s = 1, \ldots, s^* - 1, \tag{11}$$

*where $\widetilde{S}_c^s$ is the variance corrected coefficient as given in (8).*

The critical ingredient for the proof, provided in Appendix H.1, is the globally shared augmented bases. Theorem 1 bounds the drift of the low-rank representations of the local weight, which gives rise to the following global loss descent guarantee.

**Theorem 2.** *Let $U^t S^t V^{t,\top}$ and $U^{t+1} S^{t+1} V^{t+1,\top}$ be the factorization before and after iteration $t$ of Algorithm 1 with variance correction and singular value truncation threshold $\vartheta$. Let the local learning rate be $0 < \lambda \leq \frac{1}{12 L s_*}$, then the global loss descent is bounded by*

$$\mathcal{L}(U^{t+1} S^{t+1} V^{t+1,\top}) - \mathcal{L}(U^t S^t V^{t,\top}) \leq -s_* \lambda (1 - 12 s_* \lambda L) \|\nabla_{\widetilde{S}} \mathcal{L}(\widetilde{U} \widetilde{S} \widetilde{V}^\top)\|^2 + L\vartheta. \tag{12}$$

The proof is provided in Appendix H.2. The theorem shows that Algorithm 1 guarantees global loss descent, up to the error term $L\vartheta$ from low rank truncation. Further, Theorem 2 paves the way for the following result on convergence to a global stationary point.

**Theorem 3.** *Algorithm 1 guarantees that, for learning rate $\lambda \leq \frac{1}{12 L s_*}$ and final iteration $T$,*

$$\min_{t=1,\ldots,T} \left\| \nabla_{\widetilde{S}} \mathcal{L}(U^t S^t V^{t,\top}) \right\|^2 \leq \frac{48 L}{T} \left( \mathcal{L}(U^1 S^1 V^{1,\top}) - \mathcal{L}(U^{T+1} S^{T+1} V^{T+1,\top}) \right) + 48 L^2 \vartheta. \tag{13}$$

The proof is given in Appendix H.3. In particular, this theorem implies convergence of Algorithm 1 for $T \to \infty$ up to a $\vartheta$-distance to a global stationary point. This is consistent with the numerical results in Figure 1, where FedLin converges to the global minimizer (the only stationary point) while FeDLRT with variance correction stops at a point with slightly higher loss value due to a nonzero $\vartheta$. In the case that the FL problem has a low-rank solution, the truncation error bounded by $\vartheta$ vanishes, and convergence to a stationary point is guaranteed, see, e.g., Figure 3.

---

[5]Full rank $S$ is required to show consistency of the basis update step (6) with the robust operator splitting of Ceruti et al. (2022); Schotthöfer et al. (2022), see Appendix F.

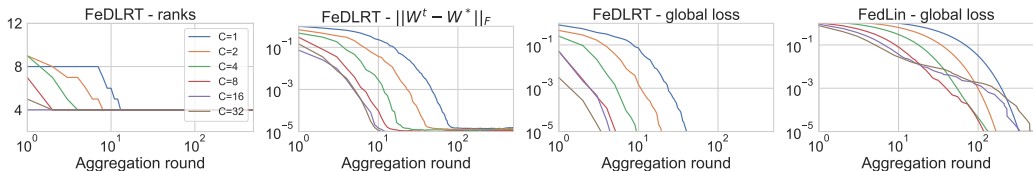

Figure 3: Comparison between FeDLRT with simplified variance correction and FedLin in the homogeneous linear least squares regression test. Each line represents the median result of 20 random initialization with $C$ clients. The plots from left to right show the rank evolution, the distance to the global optimizer, the global loss values by FeDLRT, and the global loss values by FedLin. The results show that FeDLRT converges faster in this low-rank test case by identifying (and never underestimating) the target rank $r = 4$ early in the training.

**FeDLRT convergence with simplified variance correction.** FeDLRT with simplified variance correction is detailed in Algorithm 5 with the variance correction term given in (9), which makes variance correction more communication and computation efficient but comes at a cost of the following additional assumption for convergence analysis.

**Assumption 1.** *There exists $\delta \ll 1$ such that, at each client coefficient update,*

$$\|\nabla_{\widetilde{S}}\mathcal{G}(\widetilde{U}\widetilde{S}_c^s\widetilde{V}^\top)\| - \|\nabla_S\mathcal{G}(\widetilde{U}\widetilde{S}_c^s\widetilde{V}^\top)\| < \delta\|\nabla_{\widetilde{S}}\mathcal{L}(\widetilde{U}\widetilde{S}\widetilde{V}^\top)\|, \tag{14}$$

*for functions $\mathcal{G} = \mathcal{L}$ and $\mathcal{G} = \mathcal{L}_c$, $c = 1, \ldots, C$.*

The left-hand side of (14) is the difference in the gradient dynamics induced by the basis augmentation. If FeDLRT has identified a suitable global basis $U, V$, then the small coefficient $S$ captures most of the relevant gradient information, i.e. the difference between $\nabla_{\widetilde{S}}\mathcal{G}$ and $\nabla_S\mathcal{G}$ is expected to be small. The result in Appendix D suggests that this assumption is reasonable. This scenario occurs when FeDLRT identifies the optimal rank, which could happen early for simpler problems as shown in Figure 3, or when FeDLRT approaches a stationary point.

**Theorem 4.** *Under Assumption 1, let $\mathsf{C} := s_*\lambda(1 - \delta^2 - 12s_*\lambda L + \delta^2 s_*\lambda)$. If the local learning rate $0 < \lambda \leq \frac{1}{12Ls_*}$, Algorithm 5 leads to the global loss descent*

$$\mathcal{L}(U^{t+1}S^{t+1}V^{t+1,\top}) - \mathcal{L}(U^tS^tV^{t,\top}) \leq -\mathsf{C}\|\nabla_{\widetilde{S}}\mathcal{L}(\widetilde{W}_r)\|^2 + L\vartheta.$$

The proof is provided in Appendix I.1. When $\delta$ is small, this bound is slightly weaker than the one in Theorem 2, which leads to the following corollary.

**Corollary 1.** *Assume that Assumption 1 holds. Algorithm 5 guarantees that, for the local learning rate $0 < \lambda \leq \frac{1}{s_*(12L+\delta^2)}$,*

$$\min_{t=1,\ldots,T}\left\|\nabla_{\widetilde{S}}\mathcal{L}(U^tS^tV^{t,\top})\right\|^2 \leq \frac{96L}{T}(\mathcal{L}(U^1S^1V^{1,\top}) - \mathcal{L}(U^{T+1}S^{T+1}V^{T+1,\top})) + 96L^2\vartheta.$$

The proof is analogous to the one for Theorem 3, see Appendix I.2.

### 3.3 COMPUTE AND COMMUNICATION COST

The proposed FeDLRT methods significantly reduce server and client memory footprint, the required communication bandwidth, as well as the client compute cost compared to various baselines, see Table 1. We remark that the complete federated learning process is performed on the low-rank factors, and the full matrix $W_r$ is never required, as, e.g., in Qiao et al. (2021); Xue & Lau (2023) and FeDLRT is the only low-rank method with adaptive compression incorporating variance correction, whose server compute cost scales linearly with the layer dimension since the SVD for rank truncation only needs to be computed on the augmented coefficient matrix of size $2r \times 2r$.

## 4 NUMERICAL EVALUATION

### 4.1 DISTRIBUTED LINEAR LEAST SQUARES REGRESSION

**Homogeneous test.** We first consider a (convex) FL problem (1) for linear least squares regression with local loss $\mathcal{L}_c(W) = \frac{1}{2|X_c|}\sum_{(x,y)\in X_c}\left\|p(x)^\top W p(y) - f(x,y)\right\|_2^2$, where $W \in \mathbb{R}^{n \times n}$ and

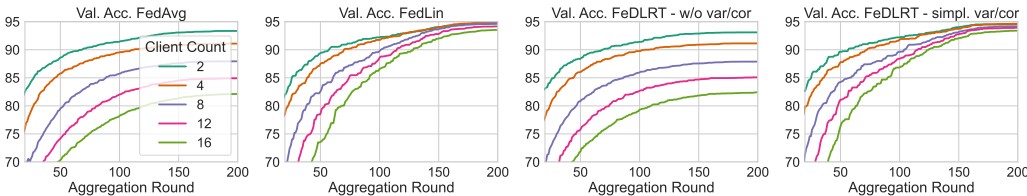

Figure 4: ResNet18 CIFAR10. In each plot, the results are reported for $C = 1, \ldots, 16$ or 32 clients with $240/C$ local iterations. We compare the convergence behavior of the median result of 10 initializations displaying the best validation accuracy until the current epoch for (from left to right) FedAvg, FedLin, FeDLRT w/o var/cor and FeDLRT w/ simplified var/cor. We observe 1) the low-rank methods closely follows the convergence dynamics of their full rank counterpart, and 2) variance correction starts to improve the convergence behavior during later stages of the training, where the non-corrected methods level off.

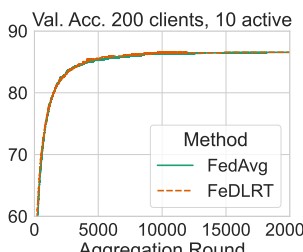

Figure 5: ResNet18 CIFAR10-i.i.d., partial participation. We compare FeDLRT w/o var. cor. to FedAvg for $C = 200$ clients, where 10 randomly sampled clients participate in each aggregation round with $s^* = 10$ local iterations. We present the median of 5 random weight initializations. The models converge to a $86.18 \pm 1.5\%$ accuracy. FeDLRT achieves a $63.32\%$ compression rate with a comparable accuracy to FedAvg.

$p : [-1, 1] \to \mathbb{R}^n$ is the Legendre polynomial basis of degree $n - 1$. The target function $f$ is manufactured as $f(x, y) = p(x)^\top W_r p(y)$, where $\text{rank}(W_r) = r$. We consider problems with $n = 20$, $r = 4$, and randomly generated $W_r$, with $10,000$ data points uniformly sampled on $[-1, 1]^2$ and uniformly distributed among clients. We compare FeDLRT with variance correction and FedLin with $s_* = 20$ local iterations and $\lambda = 1e - 3$ learning rate on $C = 1, 2, 4, 8, 16, 32$ clients. This setting satisfies the step-size restriction given in Theorem 2. In FeDLRT, the singular value truncation threshold $\vartheta = \tau ||\widetilde{S}^*||$ with $\tau = 0.1$ was used.

Figure 3 reports the dynamically updated ranks, errors, and loss values with respect to the aggregation rounds. The reported data are the medians over 20 randomly generated initial weights[6] The results indicate that FeDLRT is able to identify the correct rank within a few aggregation rounds and, furthermore, never underestimates it – which would have increased the loss value significantly. FeDLRT converges to the minimizer $W^* = W_r$ up to a $1e - 5$ error and converges faster with more clients. On this problem, FeDLRT shows up to 10x faster convergence than FedLin. We attribute this behavior to the fact that, by identifying a suitable low-rank manifold early in the training, FeDLRT significantly reduces the degrees of freedom in the FL problem.

**Heterogeneous client objective functions.** Inspired by Mitra et al. (2021a), we consider a variation of the linear least squares regression with $\mathcal{L}_c(W) = \frac{1}{2|X|} \sum_{(x,y) \in X} \left\| p(x)^\top W p(y) - f_c(x, y) \right\|^2$, where the target function $f_c$ is different for each client, and the $10,000$ training data points are available to all clients. The local target functions $f_c$ cause each client to optimize a different local problem. We choose problem size $n = 10$ with $C = 4$ clients and use learning rate $\lambda = 1e - 3$ with $s_* = 100$ local epochs. As seen in Figure 1, FeDLRT with variance correction converges (to single precision accuracy) to the minimizer $W^*$ of (1) much faster than FedLin, whereas FeDLRT without correction quickly plateaus, similar to FedAvg.

### 4.2 ResNet18 on CIFAR10

We demonstrate the performance of FeDLRT for training the exemplary ResNet18 model on CIFAR10, where we apply FeDLRT to train its fully connected head. The truncation tolerance is set to $\vartheta = \tau ||\widetilde{S}^*||$ with $\tau = 0.01$. The test case setup is summarized in Table 2. The training data is equally

---

[6]We chose to display the median trajectory to point out its convergence and monotonicity. The test case also converges in the mean.

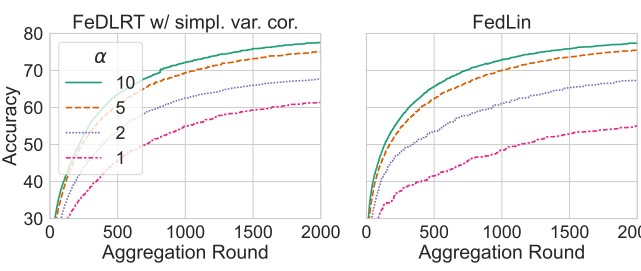

Figure 6: ResNet18 CIFAR10-Dirichlet distriubted with parameter $\alpha$ for FeDLRT w/ simpl. var. cor. and FedLin for $C = 16$ clients with $s^* = 10$ local iterations. We present the median of 5 random weight initializations. FeDLRT performance matches or surpasses the FedLin baseline.

partitioned across clients; see Appendix C.2 for the data-preprocessing details. A local iteration of Algorithm 1 at client $c$ describes one mini-batch update on the client training data set $X_c$ for a given batch size, $s_*$ is the maximum number of local iterations, and $T$ denotes the number of aggregation rounds. We see that FeDLRT ties or outperforms FedAvg in terms of final validation accuracy. Using variance correction increases the validation accuracy of FeDLRT by up to $12\%$ in this test case, matching the accuracy of FedLin and enabling FL with $93\%$ accuracy for 32 clients. For $C = 8$ clients, the communication cost saving of the compressed layers is up to $90\%$

Similar results are obtained for AlexNet, VGG16 on CIFAR10, and ViT on CIFAR100, see Appendix C, where we observe that FeDLRT closely matches the full-rank accuracy of FedLin. Lastly, we remark that variance correction ins beneficial for convergence behavior in neural network training, as shown in Figure 4.

In Figure 8 we compare the performance of full variance correction with the computationally more efficient simplified variance correction, using Algorithm 5 and observe that the latter yields similar validation accuracy, notably at higher compression ratio and communication cost reduction.

**Partial Participation:** We set the total number of clients $C = 200$, but in each aggregation round only 10 clients are randomly sampled to participate in the Algorithm 1 without variance correction for $s^* = 10$ local iterations. We show in Figure 5 that FeDLRT still mirrors the performance of FedAvg at $63.32\%$ compression rate.

**Non-i.i.d data distribution**: In the setting of Section 4.2 we explore the effect of non-i.i.d training data distributed across $C = 16$ clients on FeDLRT, where we adopt the Dirichlet distribution Hsu et al. (2019) with parameter $\alpha \in \{1, 2, 5, 10\}$. We compare the validation accuracy of FeDLRT w/ simpl. var. cor. to FedLin in Figure 6 and observe that FeDLRT matches the performance of FedLin for $\alpha = 5, 10$ and surpasses FedLin for $\alpha = 1, 2$ at compression rates between $59.8\%$ and $62.2\%$.

Figure 7: MNIST communication cost, $C = 100$ clients with 2 classes each. Comparison values are taken from (Haddadpour et al., 2020a, Figure 4).

| Method | Global Training Loss | Comm. Bits (up-link) |
|---|---|---|
| FeDLRT | 0.06 | 0.23e9 |
| SCAFFOLD | 0.05 | 1.21e9 |
| FedGATE | 0.13 | 0.61e9 |
| FedCOMGATE | 0.05 | 0.18e9 |
| FedAvg | 0.15 | 0.62e9 |
| FedPAQ | 0.15 | 0.18e9 |

**Communication cost**: We compare the effective communication cost and global training loss after 100 aggregation rounds using an MLP trained on heterogeneous MNIST data in Figure 7, see Appendix C.5 for details. FeDLRT has comparable communication cost and global training loss compared to FedCOMGATE Haddadpour et al. (2020a), the best performing reference method. We remark that FeDLRT is the only compared method with with client compute cost reduction.

**In conclusion**, we have presented FeDLRT, an efficient low-rank FL scheme with convergence guarantees and automatic compression, and demonstrated its capabilities in several test cases.

**Limitations and future work:** We remark that the underlying assumption for this work is that the target model can be expressed sufficiently well via a low-rank representation. Although the communication cost in terms of transferred parameters is significantly reduced compared to existing method, FeDLRT still requires two communication handshakes for one aggregation round, just like its full-rank counterpart FedLin. Therefore, the method needs to be refined for scenarios where the clients have different communication latencies or for completely asynchronous scenarios. Potential future research directions include performing large-scale tests with thousands of clients, extending the algorithm to incorporate other client drift mitigation techniques, e.g. Liang et al. (2019); Ivkin et al. (2020), and analyzing the convergence properties in these scenarios.

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

## A ADDITIONAL ALGORITHMS

In the following, we list a set of algorithms that are used in the paper as a contribution or as a baseline method. In particular, Algorithm 2 contains auxiliary function definitions for Algorithm 1 and Algorithm 5. Algorithm 3 is the standard FedAvg method as presented in McMahan et al. (2016). Algorithm 4 is the FedLin Algorithm Mitra et al. (2021a), i.e. the extension of Algorithm 4 with variance correction. Algorithm 5 represents the FeDLRT method with simplified variance correction, as analyzed in Theorem 4 and Corollary 1 with the additional Assumption 1.

---

**Algorithm 2:** Auxiliary functions

1 **def** broadcast($\{M_i\}_i$: *list of matrices*)**:**
2   | Send $M_i$ from server to all clients $\forall i$
3 **def** aggregate($\{M_{c,i}\}_i$: *list of matrices*)**:**
4   | Send $M_{c,i}$ from client to server $\forall c, i$
5   | $M_i \leftarrow \frac{1}{C}\sum_{c=1}^{C} M_c \quad \forall i$
6   | return $\{M_i\}_i$;
7 **def** coefficient_update_var_cor($c$: *client*, $V_c$: *correction term*)**:**
8   | **for** $s = 0, \ldots, s_* - 1$ **do**                                /* On client */
9   |   | $\widetilde{S}_c^{s+1} \leftarrow \widetilde{S}_c^s - \lambda\left(\nabla_{\widetilde{S}}\mathcal{L}_c(\widetilde{U}_c\widetilde{S}_c^s\widetilde{V}_c^\top) + V_c\right)$
10 **def** coefficient_update($c$: *client*)**:**
11   | **for** $s = 0, \ldots, s_* - 1$ **do**                                      /* On client */
12   |   | $\widetilde{S}_c^{s+1} \leftarrow \widetilde{S}_c^s - \lambda\nabla_{\widetilde{S}}\mathcal{L}_c(\widetilde{U}_c\widetilde{S}_c^s\widetilde{V}_c^\top)$
13 **def** basis_augmentation($B$: *old basis*, $G_B$: *basis dynamics*)**:**
14   | $[B \mid \bar{B}] \leftarrow$ qr($[B \mid G_B]$)                                    /* On server */
15   | return $\bar{B}$

---

**Algorithm 3:** FedAvg McMahan et al. (2016). (See Algorithm 2 for auxiliary function definitions)

**Input :** Initial values for weight matrix $W$
Client-server setup with clients $c = 1, \ldots, C$.
1 **for** $t = 1, \ldots, T$ **do**
2   | broadcast($\{W^t\}$)
3   | $W_c^{s=0} \leftarrow W^t$
4   | **for** $s = 0, \ldots, s_* - 1$ **do**
5   |   | $W_c^{s+1} \leftarrow W_c^s - \lambda\nabla_W\mathcal{L}_c(W_c^s)$      /* Gradient descent on client */
6   | $W^{t+1} \leftarrow$ aggregate($\{W_c^{s_*}\}$)            /* Aggregation on server */

---

**Algorithm 4:** FedLin Mitra et al. (2021a). (See Algorithm 2 for auxiliary function definitions)

**Input :** Initial values for weight matrix $W$
Client-server setup with clients $c = 1, \ldots, C$.
1 **for** $t = 1, \ldots, T$ **do**
2   | broadcast($\{W^t\}$)
3   | $G_{W,c} \leftarrow \nabla_W\mathcal{L}_c(W^t)$                /* Gradient computation on client */
4   | $G_W \leftarrow$ aggregate($\{G_{W,c}\}$)              /* Aggregation on server */
5   | broadcast($\{G_W\}$)
6   | $W_c^{s=0} \leftarrow W^t$
7   | $V_c \leftarrow G_W - G_{W,c}$        /* Correction term computation on client */
8   | **for** $s = 0, \ldots, s_* - 1$ **do**
9   |   | $W_c^{s+1} \leftarrow W_c^s - \lambda\nabla_W\mathcal{L}_c(W_c^s) + V_c$    /* Corrected iteration on client */
10   | $W^{t+1} \leftarrow$ aggregate($\{W_c^{s_*}\}$)            /* Aggregation on server */

---

---

**Algorithm 5:** FeDLRT with simplified variance correction. (See Algorithm 2 for auxiliary function definitions)

---

**Input:** Initial orthonormal bases $U^1, V^1 \in \mathbb{R}^{n \times r}$ and full rank $S^1 \in \mathbb{R}^{r \times r}$;
Client-server setup with clients $c = 1, \ldots, C$;
$\tau$: singular value threshold for rank truncation.

1 **for** $t = 1, \ldots, T$ **do**
2 $\quad$ broadcast$(\{U^t, V^t, S^t\})$
3 $\quad$ $G_{U,c} \leftarrow \nabla_U \mathcal{L}_c(U^t S^t V^{t,\top})$ $\hfill$ /* On client */
4 $\quad$ $G_{V,c} \leftarrow \nabla_V \mathcal{L}_c(U^t S^t V^{t,\top})$ $\hfill$ /* On client */
5 $\quad$ $G_{S,c} \leftarrow \nabla_S \mathcal{L}_c(U^t S^t V^{t,\top})$ $\hfill$ /* On client */
6 $\quad$ $G_U, G_V, G_S \leftarrow$ aggregate$(\{G_{U,c}, G_{V,c}, G_{S,c}\})$
7 $\quad$ $\bar{U} \leftarrow$ basis_augmentation$(U^t, G_U)$, $\bar{V} \leftarrow$ basis_augmentation$(V^t, G_V)$
8 $\quad$ broadcast$\big(\{\bar{U}, \bar{V}, G_S\}\big)$
9 $\quad$ $\widetilde{U} \leftarrow [U^t \mid \bar{U}], \widetilde{V} \leftarrow [V^t \mid \bar{V}]$ $\hfill$ /* Basis assembly on client */
10 $\quad$ $\widetilde{S}^{s=0} \leftarrow \begin{bmatrix} S^t & 0 \\ 0 & 0 \end{bmatrix}$ $\hfill$ /* Coefficient matrix assembly on client */
11 $\quad$ $\check{G}_{\widetilde{S},c} \leftarrow \begin{bmatrix} G_{S,c} & 0 \\ 0 & 0 \end{bmatrix}$ $\hfill$ /* Client coeff. gradient approximation on client */
12 $\quad$ $\check{G}_{\widetilde{S}} \leftarrow \begin{bmatrix} G_S & 0 \\ 0 & 0 \end{bmatrix}$ $\hfill$ /* Global coeff. gradient approximation on client */
13 $\quad$ coefficient_update_var_cor$\Big(c, \check{G}_{\widetilde{S}} - \check{G}_{\widetilde{S},c}\Big)$ $\hfill$ /* On client */
14 $\quad$ $\widetilde{S}^* \leftarrow$ aggregate$\Big(\Big\{\widetilde{S}_c^{s_*}\Big\}\Big)$
15 $\quad$ $P_{r_1}, \Sigma_{r_1}, Q_{r_1} \leftarrow$ svd$(\widetilde{S}^*)$ with threshold $\vartheta$ $\hfill$ /* Compression step */
16 $\quad$ $U^{t+1} \leftarrow \widetilde{U} P_{r_1}$, and $V^{t+1} \leftarrow \widetilde{V} Q_{r_1}$ $\hfill$ /* Basis projection */
17 $\quad$ $S^{t+1} \leftarrow \Sigma_{r_1}$

---

## B    EXTENSION TO CONVOLUTIONS AND TENSOR-VALUED WEIGHTS

FeDLRT can readily be extended to tensor-valued neural network layers, e.g. convolutional layers, following Zangrando et al. (2023), where, e.g., a 2D convolution kernel is interpreted as an order-4 tensor and factorized by using the Tucker decomposition. To this end, the Tucker bases $U_i \in \mathbb{R}^{n_i \times r_i}$ for $i = 1, 2, 3, 4$ replace the $U$ and $V$ bases in the matrix case, and the Tucker core tensor $C \in \mathbb{R}^{r_1 \times r_2 \times r_3 \times r_4}$ replaces the coefficient matrix $S$, to which the variance correction is applied. The analysis holds for the Tucker Tensor case, since Tucker Tensors have a manifold structure. In the analysis, one needs to consider the gradient projected upon all bases $U_i$ instead of $U$ and $V$. The compression step is performed with an truncated Tucker decomposition of the core tensor $C$, instead of an SVD of $S$. For intuition, one can also refer to the matrix case as the order-2 Tucker Tensor case. Remark that the bases $U_i$ are all updated simultaneously, thus the adaption to the tensor case does not require more communication rounds.

## C    ADDITIONAL NUMERICAL EVALUATION

### C.1    COMPUTE RESOURCES

The convex test cases are computed on a single Nvidia RTX 4090 GPU. The computer vision benchmarks use a set of Nvidia Tesla V100-SXM2-16GB and Tesla P100-PCIE-16GB. For prototyping, a Nvidia GTX1080ti is used.

---

**Algorithm 6:** Naive implementation of FeDLRT. (See Algorithm 2 for auxiliary function definitions)

---

**Input :** Initial orthonormal bases $U^1, V^1 \in \mathbb{R}^{n \times r}$ and full rank $S^1 \in \mathbb{R}^{r \times r}$;
Client-server setup with clients $c = 1, \ldots, C$;
$\tau$: singular value threshold for rank truncation.

**1 for** $t = 1, \ldots, T$ **do**
  **2**    $\texttt{broadcast}(\{U^t, V^t, S^t\})$
  **3**    $U_c^{s=0}, V_c^{s=0}, S_c^{s=0} \leftarrow U^t, V^t, S^t$
  **4**    **for** $s = 0, \ldots, s_* - 1$ **do**                      /* On client */
    **5**      $G_{U,c} \leftarrow \nabla_U \mathcal{L}_c(U_c^s S_c^s V_c^{s,\top})$
    **6**      $G_{V,c} \leftarrow \nabla_V \mathcal{L}_c(U_c^s S_c^s V_c^{s,\top})$
    **7**      $\widetilde{U}_{c,\_} \leftarrow \texttt{qr}([U_c^s \mid G_{U,c}])$
    **8**      $\widetilde{V}_{c,\_} \leftarrow \texttt{qr}([V_c^s \mid G_{V,c}])$
    **9**      $\widetilde{S}_c = \widetilde{U}_c^\top U_c^s S_c^s V_c^{s,\top} \widetilde{V}_c$
    **10**      $\widetilde{S}_c^* \leftarrow \widetilde{S}_c - \lambda \nabla_{\widetilde{S}} \mathcal{L}_c(\widetilde{U}_c \widetilde{S}_c \widetilde{V}_c^\top)$
    **11**      $\widetilde{S}^* \leftarrow \texttt{aggregate}\left(\left\{\widetilde{S}_c^*\right\}\right)$
    **12**      $P_{r_1}, \Sigma_{r_1}, Q_{r_1} \leftarrow \texttt{svd}(\widetilde{S}^*)$ with threshold $\vartheta$      /* Compression step */
  **13**    $U^{t+1} \leftarrow \widetilde{U} P_{r_1}$, and $V^{t+1} \leftarrow \widetilde{V} Q_{r_1}$      /* Basis projection */
  **14**    $S^{t+1} \leftarrow \Sigma_{r_1}$

---

## C.2   DATA AUGMENTATION

We use standard data augmentation techniques for the proposed test cases. That is, for CIFAR10, we augment the training data set by a random horizontal flip of the image, followed by a normalization using mean $[0.4914, 0.4822, 0.4465]$ and std. dev. $[0.2470, 0.2435, 0.2616]$. The test data set is only normalized. The same augmentation is performed for CIFAR100, where with mean $[0.5071, 0.4867, 0.4408]$ and std. dev. $[0.2673, 0.2564, 0.2762]$.

## C.3   ADDITIONAL COMPUTER VISION RESULTS

**ResNet18 on CIFAR10:** We provide a comparison of the full variance correction and simplified variance correction-based FeDLRT method with ResNet18 on Cifar10. We display the statistics for 10 random initializations; each warm-started with 5 central learning iterations. We set $s_* = 240/C$ so that in each training run, the global network iterates through the same amount of data. This setup favors low client counts, and, as expected, the validation accuracy drops as $C$ grows for FedAvg and FeDLRT without variance correction, see Figure 4. Figure 8 shows that both variants perform equally well.

**AlexNet on CIFAR10:** We train AlexNet on CIFAR10, where the fully connected head of the network is replaced by a low-rank counterpart. A federated neural network setup with $C$ clients trains on $CTs_*$ random batches of the dataset, that is the number of seen training data batches scales with the client count. Figure 9 displays the validation accuracy of FeDLRT with variance correction compared to FedLin, where one can see that the performance of FeDLRT mirrors the performance of FedLin with more degrees of freedom. The measured validation accuracy peaks at $C = 4$ clients in both cases, where the higher number of seen training data-points offsets the negative effects of more clients on the validation performance. All reported runs are within close distance of the non-federated, full-rank baseline accuracy of $85.6\%$. Communication cost savings of the fully connected layers amount between $96\%$ and $97\%$ [7] We observe that, similarly to the results in Section 4.1, the maximum achieved communication cost savings, which depend on the layer ranks scales with the number of clients $C = 4$, indicating that the decay rate of the singular values of the averaged coefficient matrix $\widetilde{S}^*$ depends on $C$.

---

[7] For clarity of exposition we consider only the fully connected layers. Taking into account the non low-rank convolution layers, the communication cost savings reduces to $87.5\%$ to $87.3\%$.

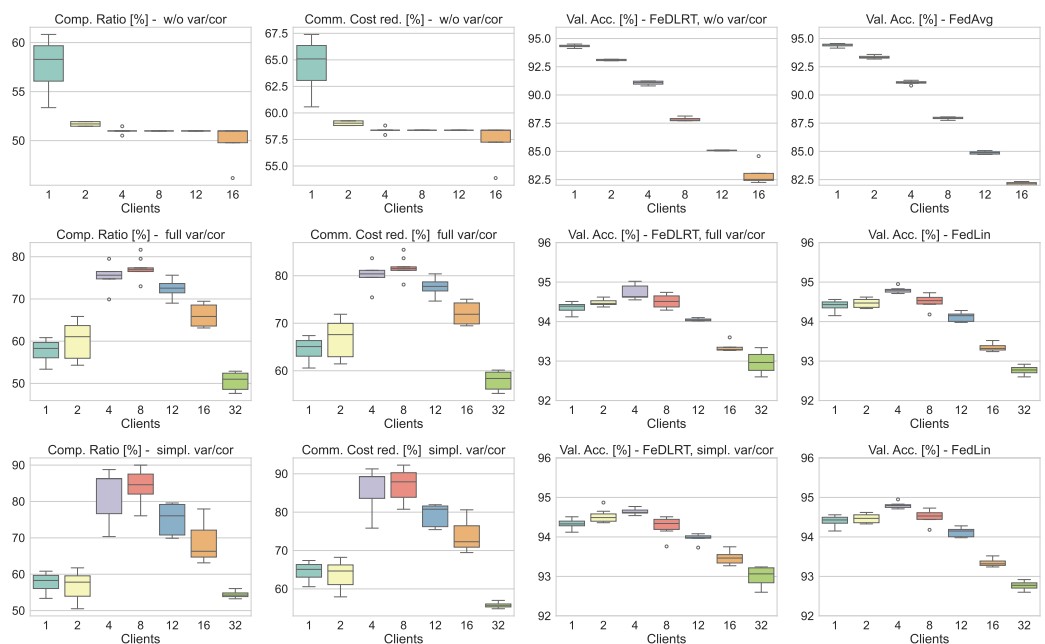

Figure 8: Comparisons for training ResNet18 on CIFAR10 benchmark. Top row compares FeDLRT without variance correction to FedAvg, middle and bottom rows compare FeDLRT with full and simplified variance correction to FedLin, respectively. In each row, the left two panels show the model compression ratio and the communication cost reduction from FeDLRT, and the right two panels show the validation accuracy for FeDLRT and the full-rank counterparts. In each plot, the results are reported for $C = 1, \ldots, 16$ or 32 clients with $240/C$ local iterations. FeDLRT matches the accuracy of FedAvg and FedLin well, while substantially reducing the server and client memory and communication costs. Variance correction leads to an up to $12\%$ increase in validation accuracy for large $C$, mitigating the client drift problem. The simplified variance correction (bottom row) gives comparable results to full version (middle row) at a lower communication and computation cost.

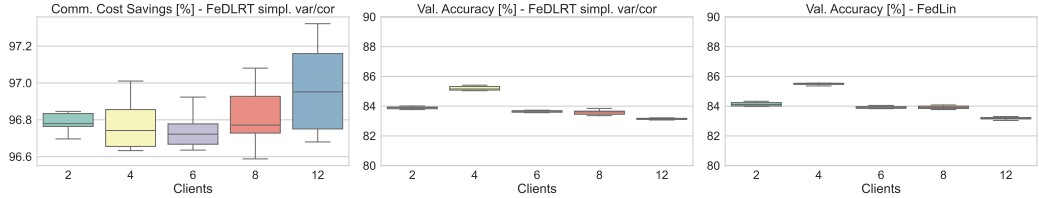

Figure 9: AlexNet CIFAR10 benchmark with fixed number of local iterations. (Left Panel) shows the savings in communication cost of simplified variance corrected FeDLRT vs FedLin. (Mid and right panel) compares the validation accuracy of FeDLRT and FedLin, where we see that FeDLRT behaves similarly to FedLin and achieves accuracy levels near the non-federated baseline value of $85.6\%$.

**VGG16 on CIFAR10:** We train AlexNet on CIFAR10, where the fully connected head of the network is replaced by a low-rank counterpart. A federated neural network setup with $240/C$ local iterations for $C$ clients. Figure 10 displays the validation accuracy of FeDLRT with variance correction compared to FedLin, where one can see that the performance of FeDLRT mirrors the performance of FedLin with more degrees of freedom. All reported runs are within close distance of the non-federated, full-rank baseline accuracy of $85.6\%$. Communication cost savings of the fully

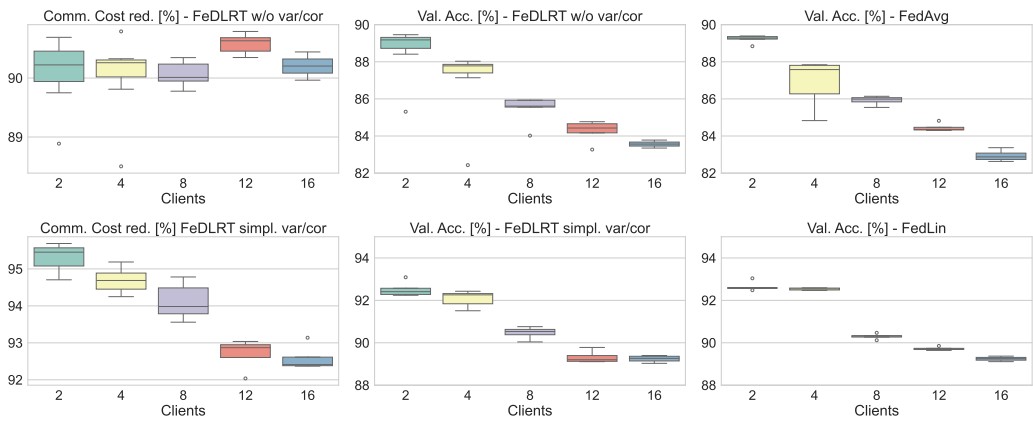

Figure 10: VGG16 CIFAR10 benchmark with $240/C$ local iterations for $C$ clients with simplified (lower row) and without (upper row) variance correction. (Left panel) show the savings in communication cost corresponding to FedLin at final time. (Mid and right panel top row) compares the validation accuracy of FeDLRT and FedAvg, where we see that FeDLRT behaves similarly to FedAvg, where higher $C$ correlates with a drop in accuracy. FeDLRT with variance correction mitigates this issue and achieves similar performance as FedLin, close to the non-federated baseline accuracy is $93.15\%$.

connected layers amount between $96\%$ and $97\%$ [8] We observe, similarly results as in the ResNet18 test case.

**VGG16 on CIFAR10 with low-rank convolutions:** Mirroring the compute setup of the VGG16 test-case above, we now rewrite all convolutional layers of VGG16 as order 4 tensors in low-rank Tucker format, as described in appendix B. The full-connected head of the network is treated with the matrix low-rank method. The corresponding training results can be seen in Figure 11, and correspond well with the previous results for VGG16. The reduction of communication cost is slightly higher, due to the compression of the convolutions.

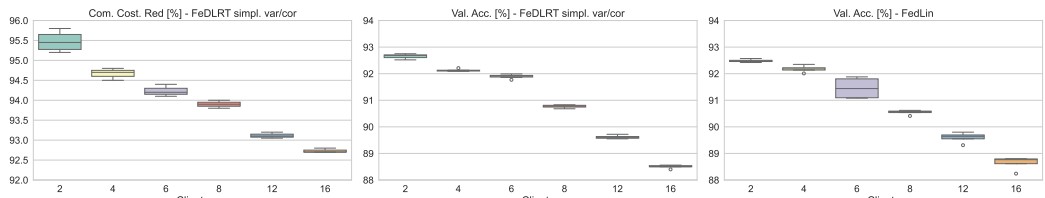

Figure 11: VGG16 CIFAR10, low-rank convolutional layers and low-rank fully connected layers. We report the communication cost savings and the validation accuracy of VGG16 with FeDLRT applied to training convolution and classifier layers. 2D convolutions are interpreted as an order-4 tensor and factorized in the Tucker format. The statistics over five random network initializations are reported using the training hyperparemeters of Table 2 of the main manuscript. The results are similar to Fig. 7 in the main manuscript, where only the classifier is compressed. Remark that here the classifier contains most of the network parameters.

**Vision Transformer on CIFAR100:** We consider a small vision transformer for CIFAR100, with 6 attention layers with 2 heads each followed by a ResNet block and a drop-out layer, all with weight matrices of dimension $512 \times 512$. The tokenizer takes patches of size 8 with embedding dimension 512. Training hyperparameters are given in Table 2. Remark that we do not aim for SOTA performance, since transformer architectures are notoriously difficult to compress with low-rank

---

[8]For clarity of exposition we consider only the fully connected layers. Taking into account the non low-rank convolution layers, the communication cost savings reduces to $87.5\%$ to $87.3\%$.

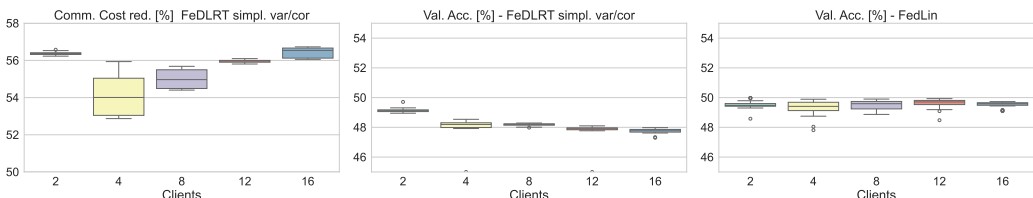

Figure 12: ViT CIFAR100 benchmark. (Left Panel) shows the savings in communication cost of variance corrected FeDLRT vs FedLin. (Mid and right panel) compares the validation accuracy of FeDLRT and FedLin, where we see that FeDLRT behaves similarly to FedLin and achieves accuracy levels near the non-federated baseline value of $50\%$, which is similar to literature results Zhu et al. (2023).

Table 2: Experimental setup object detection benchmarks. All test cases use a cosine annealing learning rate scheduler.

|  | Alexnet/Cifar10 | ResNet18/Cifar10 | VGG16/Cifar10 | ViT/Cifar100 |
|---|---|---|---|---|
| Batch size | 128 | 128 | 128 | 256 |
| Start Learningrate | 1e−2 | 1e−3 | 1e−2 | 3e−4 |
| End Learningrate | 1e−5 | 5e−4 | 5e−4 | 1e−5 |
| Aggregation Rounds | 200 | 200 | 200 | 200 |
| Local Iterations | 100 | 240/C | 240/C | 240/C |
| Truncation tolerance $\tau$ | 0.01 | 0.01 | 0.01 | 0.01 |
| Momentum | 0.0 | 0.9 | 0.1 | n.a. |
| Weight Decay | 1e−4 | 1e−3 | 1e−4 | 1e−2 |
| Optimizer | SGD | SGD | SGD | Adam w/ std pytorch parameters |

approaches, but rather compare the performance of FedLin to FeDLRT for a given compute budget. We use $s_* = 240/C$ local iterations for $C$ clients. Observe in Figure 12 that FeDLRT achieves similar performance as ViT with over $55\%$ communication cost savings on average.

## C.4 COMPUTE COST ILLUSTRATION

We illustrate the compute and communication cost for FeDLRT in comparison to other low-rank methods in Figure 13.

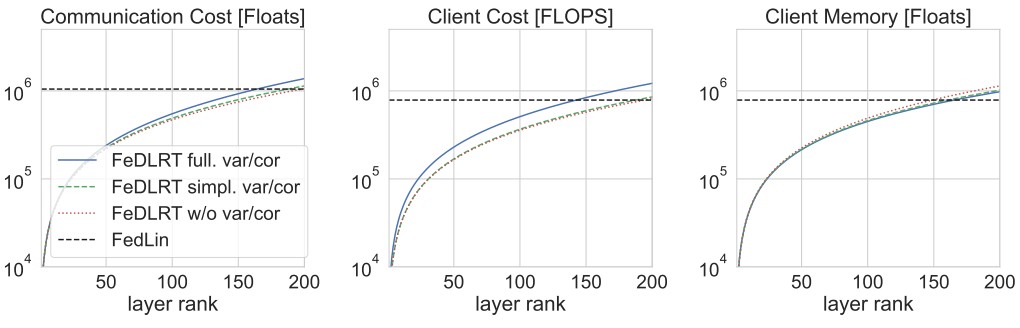

Figure 13: Scaling of communication cost (left) compute cost at a single client (middle), and client memory footprint (right) for $s_* = 1$ client iteration and a single data-point for $W \in \mathbb{R}^{n \times n}$ with $n = 512$. In practice we have $r \ll n$, see Section 4.

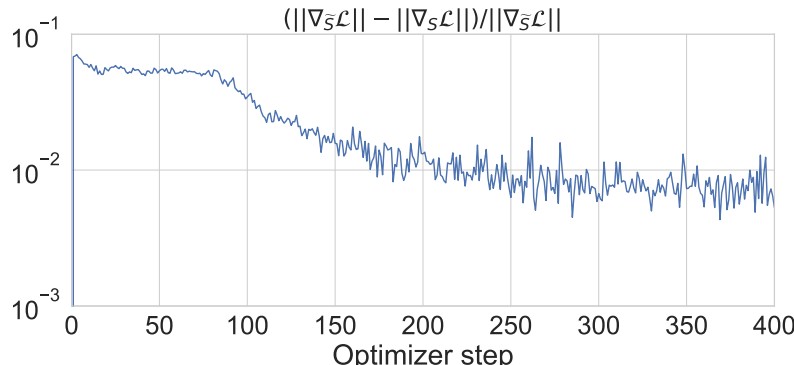

Figure 14: Evaluation of the gradients in Assumption 1 for ResNet18 on Cifar10 with i.i.d data and $C = 10$ clients with the settings of Table 2.

### C.5 COMMUNICATION COST COMPARISON

We compare the communication cost of FeDLRT w/ simpl. var. cor. to SCAFFOLD Karimireddy et al. (2020), FedGate, FedCOMGATE Haddadpour et al. (2020a), FedAvg, and FedPAQ Reisizadeh et al. (2020a) in the MNIST benchmark with heterogeneous data such that each of the $C = 100$ clients have two classes available. We use the MLP of (Haddadpour et al., 2020a, Section 6) with 200 neurons and 2 hidden layers as defined in https://github.com/MLOPTPSU/FedTorch where we replace the hidden and input layer with low-rank layers. We train for 100 iterations with hyper-parameters obtained by a random search: learning rate $\lambda = 0.009$, 20 local iterations, $\tau = 0.11$, batch size 128 and display the results in Figure 7. FeDLRT has competitive communication costs. We remark that FedCOMGATE, SCAFFOLD and FedPAQ train on the full weight matrix on the client thus have higher client compute cost.

## D EMPIRICAL EVALUATION OF ASSUMPTION 1

We consider ResNet18 on Cifar10 with i.i.d data and $C = 10$ clients with the settings of Table 2. We consider client $c = 1$ and plot the term

$$\frac{\|\nabla_{\widetilde{S}}\mathcal{G}(\widetilde{U}\widetilde{S}_c^s\widetilde{V}^\top)\| - \|\nabla_S\mathcal{G}(\widetilde{U}\widetilde{S}_c^s\widetilde{V}^\top)\|}{\|\nabla_{\widetilde{S}}\mathcal{L}(\widetilde{U}\widetilde{S}\widetilde{V}^\top)\|}$$

which should be smaller than $\delta$ by Assumption 1 for 400 optimization steps of Algorithm 1 in fig. 14. As seen, $\delta < 0.01$ for the last 200 iterations and we always have $\delta < 0.1$, thus the assumption is fulfilled.

## E NOTATION OVERVIEW FOR THE NUMERICAL ANALYSIS

We establish a set of notations to simplify the notation in the proofs

- $\mathcal{L}_c(W)$ denotes the local loss function based on dataset $X_c$ at client $c$.
- $\mathcal{L}(W) = \frac{1}{C}\sum_{c=1}^{C}\mathcal{L}_c(W)$ is the global loss function.
- $F_c(W) = -\nabla_W\mathcal{L}_c(W)$ is the negate of local loss gradient.
- $F(W) = \frac{1}{C}\sum_{c=1}^{C}F_c(W)$ is the negate of global loss gradient.
- $\mathcal{M}_r = \{W \in \mathbb{R}^{n \times n} : \text{rank}(W) = r\}$ is a manifold of rank $r$ matrices.
- $W_r = USV^\top \in \mathcal{M}_r$ is a rank-$r$ approximation of a matrix $W$.
- $\mathcal{T}_{W_r}\mathcal{M}_r$ is the tangent space of $\mathcal{M}_r$ at $W_r$.
- $P(W_r)$ is the orthogonal projection onto $\mathcal{T}_{W_r}\mathcal{M}_r$.
- $P_U = UU^\top$ is the orthogonal projection onto the range of orthonormal $U \in \mathbb{R}^{n \times r}$.
- $P_V = VV^\top$ is the orthogonal projection onto the range of orthonormal $V \in \mathbb{R}^{n \times r}$.

- When applied to vectors, $\|\cdot\|$ denotes the Euclidean norm ($\ell_2$-norm). When applied to matrices, $\|\cdot\|$ denotes the Frobenius norm.

L- smoothness in this paper is defined as L-continuity of a function and its gradient, where both L constants are identical. That is for a function $f(x)$ with gradient $G(x)$, we have $\|f(x_1) - f(x_2)\| \le L\|x_1 - x_2\|$ and $\|G(x_1) - G(x_2)\| \le L\|x_1 - x_2\|$ for some $L > 0$.

## F  EFFICIENT BASIS GRADIENT DYNAMICS FOR BASIS AUGMENTATION

We first consider the basis update & Galerkin splitting scheme of (5). The splitting performs a reparametrization of the form $K(t) = U(t)S(t)$ and $L(t) = V(t)S(t)^\top$. The basis update then reads

$$\dot{K} = -\nabla_K \mathcal{L}(K(t)V_0^\top) \in \mathbb{R}^{n \times r}, \quad K(0) = U_0 S_0,$$
$$\dot{L} = -\nabla_L \mathcal{L}(U_0 L(t)^\top) \in \mathbb{R}^{n \times r}, \quad L(0) = V_0 S_0^\top. \tag{15}$$

Given the solution $K(t_1)$ and $L(t_1)$ at time $t_1$, the bases $U_0$ and $V_0$ are augmented by the orthonormalization of the new directions $K(t_1)$ and $L(t_1)$, i.e.

$$\widetilde{U}R = \mathtt{qr}([U_0 \mid K(t_1)]) \in \mathbb{R}^{n \times 2r},$$
$$\text{and} \quad \widetilde{V}R = \mathtt{qr}([V_0 \mid L(t_1)]) \in \mathbb{R}^{n \times 2r}, \tag{16}$$

where $R$ is the right factor of the respective QR decomposition and can be discarded. The initial condition of the coefficient update is $S(t_0)$ projected onto the new bases, i.e.,

$$\dot{\widetilde{S}} = -\nabla_S \mathcal{L}(\widetilde{U}\widetilde{S}(t)\widetilde{V}^\top), \quad \widetilde{S}(0) = \widetilde{U}^\top U_0 \widetilde{S}(0) V_0^\top \widetilde{V}. \tag{17}$$

After the integration of the coefficient dynamics above, the redundant basis functions are typically truncated via an SVD of $S$ ensuring that $S$ is always full rank. In its continuous form above, the splitting yields a robust integrator for the projected gradient flow, without manifold dependent step-size restrictions:

**Theorem 5.** *(Schotthöfer et al. (2022)) Assume $\mathcal{L}$ is L-smooth with constant $L$, and locally bounded by $B$. Let $W_r(t)$ be the low-rank continuous time solution of (15) and (17) and let $W(t)$ be the full rank solution at $t = 0$. Assume the $K, L,$ and $S$ equations are integrated exactly from time $t = 0$ to $\Delta t$. Assume that for any $Y \in \mathcal{M}_r$ sufficiently close to $W_r(t)$ the gradient $F(Y)$ is $\epsilon$ close to $\mathcal{M}_r$. Then*

$$\|W(\Delta t) - W_r(\Delta t)\| \le d_1\epsilon + d_2\Delta t + d_3\frac{\vartheta}{\Delta t},$$

*where $d_1, d_2, d_3$ depend only on $L$ and $B$.*

The theorem guarantees that the low-rank representation does not imply any step-size restrictions on the optimization scheme. This is in stark contrast to a naive alternating descent optimization of the low-rank factors $U, S, V$.

To build an discretized numerical optimizer in a resource constrained federated scenario from the above continuous splitting equations, we avoid the reparametrization, which implies a 200% memory cost increase on the client side, since three versions of the low-rank layer need to be tracked.

**Lemma 2.** *Let $USV \in \mathcal{M}_r$ be a low rank factorization that follows the projected gradient (5) flow using the splitting scheme (15) with $K = US$ and $V = VS^\top$. Further, assume that equations for the $K$ and $L$ factors are solved by an explicit Euler time integration with learning rate $\lambda$, i.e.*

$$K(t_1) = K(0) - \lambda\nabla_K \mathcal{L}(K(0)V_0^\top), \quad K(0) = U_0 S_0,$$
$$L(t_1) = L(0) - \lambda\nabla_L \mathcal{L}(U_0 L(0)^\top), \quad L(0) = V_0 S_0^\top. \tag{18}$$

*Then, the basis augmentation (16) can be expressed as*

$$\widetilde{U}R = \mathtt{qr}([U_0 \mid -\nabla_U \mathcal{L}(U_0 S_0 V_0^\top)]) \in \mathbb{R}^{n \times 2r},$$
$$\text{and} \quad \widetilde{V}R = \mathtt{qr}([V_0 \mid -\nabla_V \mathcal{L}(U_0 S_0 V_0^\top)]) \in \mathbb{R}^{n \times 2r}. \tag{19}$$

*and maintains the structure of the basis update and Galerkin operator split.*

*Proof.* We consider the proof for the $K$ equation and the $U$ basis; the proof for $L$ and $V$ follows analogously.

Considering (16), we obtain with the explicit Euler discretization (18),

$$
\begin{aligned}
\mathrm{span}\left([U_0 \mid K(t_1)]\right) &= \mathrm{span}\left([U_0 \mid U_0 - \lambda \nabla_K \mathcal{L}(K(0)V_0^\top)]\right) \\
&= \mathrm{span}\left([U_0 \mid -\lambda \nabla_K \mathcal{L}(K(0)V_0^\top)]\right) \\
&= \mathrm{span}\left([U_0 \mid -\nabla_K \mathcal{L}(K(0)V_0^\top)]\right).
\end{aligned}
\tag{20}
$$

Next, consider the continuous time dynamics of $\dot{K}$, where we omit explicit time dependence on $U, S, V$ and $K$ for the sake of brevity, i.e.,

$$
\begin{aligned}
\dot{K} &= (\dot{US}) \\
&= \dot{U}S + U\dot{S} \\
&\overset{(5)}{=} -(I - UU^\top)\nabla_W \mathcal{L}(USV^\top)VS^{-1}S - UU^\top \nabla_W \mathcal{L}(USV^\top)V \\
&= -(I - P_U)\nabla_W \mathcal{L}(USV^\top)V - P_U \nabla_W \mathcal{L}(USV^\top)V \\
&= (P_U - I)\nabla_W \mathcal{L}(USV^\top)V - P_U \nabla_W \mathcal{L}(USV^\top)V \\
&= -\nabla_W \mathcal{L}(USV^\top)V
\end{aligned}
\tag{21}
$$

Further, using the chain rule, we observe

$$
\nabla_U \mathcal{L}(USV^\top) = \nabla_W \mathcal{L}(USV^\top)\nabla_U(USV^\top) = \nabla_W \mathcal{L}(USV^\top)VS^\top
$$

Thus, $-\nabla_U \mathcal{L}(USV^\top)S^{-\top} = -\nabla_W \mathcal{L}(USV^\top)V = \dot{K}$. Full rankness of $S$ and (21) yield that $\mathrm{span}(-\nabla_U \mathcal{L}(USV^\top)) = \mathrm{span}(\dot{K})$. Together with (20) this yields the proof. $\square$

Lemma 2 adopts a more general result for Tucker tensors in an unpublished manuscript and simplifies the analysis for the matrix case considered here.

## G  EFFICIENT BASIS AND COEFFICIENT COMMUNICATION

Note that we have by orthogonality of the bases $\widetilde{U} = [U, \bar{U}]$ with $\bar{U} \in \mathbb{R}^{n \times r}$ and $\bar{U}^\top U = 0$ and $\widetilde{V} = [V, \bar{V}]$ with $\bar{V} \in \mathbb{R}^{n \times r}$ and $\bar{V}^\top V = 0$.

*Proof.* (Lemma 1) The basis augmented basis $[U, G_U]$ before orthonormalization already contains the orthonormal vectors given by the columns of $U$. A QR decomposition therefor only rearranges the columns of $G_U$ such that $\widetilde{U} = [U, \bar{U}]$ with $\bar{U} \in \mathbb{R}^{n \times r}$ and $\bar{U}^\top U = 0$. The analogous result holds for $\widetilde{V} = [V, \bar{V}]$. The projection onto the augmented basis therefore reads

$$
\widetilde{U}^\top U = \begin{bmatrix} U^\top U \\ \bar{U}^\top U \end{bmatrix} = \begin{bmatrix} I \\ 0 \end{bmatrix} \quad \text{and} \quad \widetilde{V}^\top V = \begin{bmatrix} V^\top V \\ \bar{V}^\top V \end{bmatrix} = \begin{bmatrix} I \\ 0 \end{bmatrix}.
\tag{22}
$$

Consequently, the augmented coefficient matrix takes the form

$$
\widetilde{S} = \widetilde{U}^\top U S V^\top \widetilde{V} = \begin{bmatrix} S & 0 \\ 0 & 0 \end{bmatrix}.
\tag{23}
$$

$\square$

## H  ANALYSIS FOR FEDLRT WITH FULL VARIANCE CORRECTION

In this section we establish bounds on the coefficient drift of the FeDLRT method with full variance correction. We use the established coefficient drift bound to derive a loss-descent guarantee. The strategy of our analysis follows the one of FedLin Mitra et al. (2021a). We first state an auxiliary lemma.

**Lemma 3.** *Let $U \in \mathbb{R}^{n \times r}$ and $V \in \mathbb{R}^{n \times r}$ be orthonormal matrices. Let $F$ be an $L$-continuous function. Then, for $S_1, S_2 \in \mathbb{R}^{r \times r}$,*

$$\left\| P_U \left( F(US_1V^\top) - F(US_2V^\top) \right) P_V \right\| \le L \left\| S_1 - S_2 \right\| \tag{24}$$

*and*

$$\left\| U \left( F(US_1V^\top) - F(US_2V^\top) \right) V^\top \right\| \le L \left\| S_1 - S_2 \right\|, \tag{25}$$

*where $P_U$ and $P_V$ are orthogonal projections defined in Appendix E.*

*Proof.* For the first statement, consider

$$
\begin{aligned}
&\left\| P_U \left( F(US_1V^\top) - F(US_2V^\top) \right) P_V \right\| \\
&= \left\| UU^\top \left( F(US_1V^\top) - F(US_2V^\top) \right) VV^\top \right\| \\
&\overset{(I)}{\le} \|U\| \|U^\top\| \left\| F(US_1V^\top) - F(US_2V^\top) \right\| \|V\| \|V^\top\| \\
&\overset{(II)}{=} \left\| F(US_1V^\top) - F(US_2V^\top) \right\| \\
&\overset{(III)}{\le} L \left\| US_1V^\top - US_2V^\top \right\| = L \left\| U(S_1 - S_2)V^\top \right\| \\
&\overset{(I)}{\le} L \|U\| \|S_1 - S_2\| \|V^\top\| \\
&\overset{(II)}{=} L \|S_1 - S_2\|,
\end{aligned}
$$

where we have used in (I) the operator norm inequality of the Frobenius norm, in (II) orthonormality of $U, V$, and in (III) $L$-continuity of $F$. The second statement is proven analogously. $\square$

### H.1 COEFFICIENT DRIFT BOUND FOR FEDLRT WITH FULL VARIANCE CORRECTION

We consider the FeDLRT method with variance correction, see Algorithm 1. Key difference to the FeDLRT method without variance correction is the modified coefficient update, incorporating global gradient information of the augmented coefficient matrix $\widetilde{S}$ and local, stale gradient information of the augmented coefficient matrix $\widetilde{S}_c$. The variance corrected local coefficient update (8) can be expressed in terms of the projected Riemannian gradient as

$$\widetilde{S}_c^{s+1} = \widetilde{S}_c^s + \lambda \widetilde{U}^\top \left( F_c(\widetilde{W}_{r,c}^s) - F_c(\widetilde{W}_r) + F(\widetilde{W}_r) \right) \widetilde{V}, \tag{26}$$

where $\widetilde{U}^\top F_c(\widetilde{W}_{r,c}^s)\widetilde{V} = \nabla_{\widetilde{S}_c} \mathcal{L}_c(\widetilde{U}\widetilde{S}_c^s\widetilde{V})$, $\widetilde{U}^\top F_c(\widetilde{W}_{r,c})\widetilde{V} = \nabla_{\widetilde{S}_c} \mathcal{L}_c(\widetilde{U}\widetilde{S}_c^{s=0}\widetilde{V})$ and $\widetilde{U}^\top F_c(\widetilde{W}_{r,c}^s)\widetilde{V} = \nabla_{\widetilde{S}_c} \mathcal{L}(\widetilde{U}\widetilde{S}_c^s\widetilde{V})$. Recall that $\widetilde{S} = \widetilde{S}_c$ for $s = 0$.

We provide proof for Theorem 1 to bound the drift term $\left\| \widetilde{S}_c^s - \widetilde{S}_c \right\|$. We restate this theorem to the Riemannian notation and restate it below.

**Theorem 6.** *(Restatement of Theorem 1) Given augmented basis and coefficient matrices $\widetilde{U}$, $\widetilde{V}$, and $\widetilde{S}$, and $\widetilde{W}_r = \widetilde{U}\widetilde{S}\widetilde{V}^\top$. If the local learning rate $0 < \lambda \le \frac{1}{Ls_*}$ with $s_* \ge 1$ the number of local steps, for all clients $c$,*

$$\|\widetilde{S}_c^s - \widetilde{S}_c\| \le \exp(1)s_*\lambda \left\| \widetilde{U}^\top F(\widetilde{W}_r)\widetilde{V} \right\|, \quad \text{for} \quad s = 1, \dots, s^* - 1, \tag{27}$$

*where $\widetilde{S}_c^s$ is the variance corrected coefficient as given in (8).*

*Proof.* From the adjusted coefficient update in (26), we get

$$\left\| \widetilde{S}_c^{s+1} - \widetilde{S}_c \right\| = \left\| \widetilde{S}_c^s - \widetilde{S}_c + \lambda \widetilde{U}^\top \left( F_c(\widetilde{W}_{r,c}^s) - F_c(\widetilde{W}_r) + F(\widetilde{W}_r) \right) \widetilde{V} \right\|$$

$$\leq \left\| \widetilde{S}_c^s - \widetilde{S}_c \right\| + \lambda \left\| \widetilde{U}^\top \left( F_c(\widetilde{W}_{r,c}^s) - F_c(\widetilde{W}_r) \right) \widetilde{V} \right\| + \lambda \left\| \widetilde{U}^\top F(\widetilde{W}_r)\widetilde{V} \right\|$$

$$\overset{(I)}{\leq} \left\| \widetilde{S}_c^s - \widetilde{S}_c \right\| + \lambda L \left\| \widetilde{S}_c^s - \widetilde{S} \right\| + \lambda \left\| \widetilde{U}^\top F(\widetilde{W}_r)\widetilde{V} \right\|$$

$$\leq (1 + \lambda L) \left\| \widetilde{S}_c^s - \widetilde{S} \right\| + \lambda \left\| \widetilde{U}^\top F(\widetilde{W}_r)\widetilde{V} \right\|$$

$$\leq \left( 1 + \frac{1}{s_*} \right) \left\| \widetilde{S}_c^s - \widetilde{S} \right\| + \lambda \left\| \widetilde{U}^\top F(\widetilde{W}_r)\widetilde{V} \right\|.$$

We use in (I) Lemma 3 Recursively plugging in the above inequality yields for $a = (1 + \frac{1}{s_*})$

$$\left\| \widetilde{S}_c^{s+1} - \widetilde{S}_c \right\| \leq a^{s+1} \left\| \widetilde{S}_c^{s=0} - \widetilde{S} \right\| + \left( \sum_{j=0}^{s} a^j \right) \lambda \left\| \widetilde{U}^\top F(\widetilde{W}_r)\widetilde{V} \right\|$$

$$= \left( \sum_{j=0}^{s} a^j \right) \lambda \left\| \widetilde{U}^\top F(\widetilde{W}_r)\widetilde{V} \right\|$$

$$= \frac{a^{s+1} - 1}{a - 1} \lambda \left\| \widetilde{U}^\top F(\widetilde{W}_r)\widetilde{V} \right\|$$

$$\leq \left( 1 + \frac{1}{s_*} \right)^{s+1} s_* \lambda \left\| \widetilde{U}^\top F(\widetilde{W}_r)\widetilde{V} \right\|$$

$$\leq \left( 1 + \frac{1}{s_*} \right)^{s_*} s_* \lambda \left\| \widetilde{U}^\top F(\widetilde{W}_r)\widetilde{V} \right\|$$

$$\leq \exp(1) s_* \lambda \left\| \widetilde{U}^\top F(\widetilde{W}_r)\widetilde{V} \right\|.$$

$\square$

### H.2 GLOBAL LOSS DESCEND FOR FEDLRT WITH FULL VARIANCE CORRECTION

We first state a few auxiliary lemmas, which provide common inequalities that will be used in the following analysis.

**Lemma 4.** *((Hnatiuk et al., 2024, Lemma 5.2)) For any two matrices $Y_1, Y_2 \in \mathbb{R}^{n \times n}$ and an $L$-smooth $\mathcal{L}$ with constant $L$ it holds*

$$\mathcal{L}(Y_1) - \mathcal{L}(Y_2) \leq - \langle Y_1 - Y_2, F(Y_2) \rangle + \frac{L}{2} \| Y_1 - Y_2 \|^2, \tag{28}$$

*where $F(Y) = -\nabla_Y \mathcal{L}(Y)$.*

**Lemma 5.** *((Mitra et al., 2021b, Lemma 5)) For two vectors $x_1, x_2 \in \mathbb{R}^d$ it holds for $\gamma > 0$*

$$\| x_1 + x_2 \|^2 \leq (1 + \gamma) \| x_1 \|^2 + \left( 1 + \frac{1}{\gamma} \right) \| x_2 \|^2. \tag{29}$$

**Lemma 6.** *((Mitra et al., 2021b, Lemma 6)) For $C$ vectors $x_1, \ldots, x_C \in \mathbb{R}^d$ the application of Jensen's inequality yields*

$$\left\| \sum_{c=1}^{C} x_c \right\|^2 \leq C \sum_{c=1}^{C} \| x_c \|^2. \tag{30}$$

First, we consider the loss function value at the augmentation step.

**Lemma 7.** *We have $\mathcal{L}(\widetilde{W}_r) = \mathcal{L}(W_r^t)$ for the loss before and after basis augmentation.*

*Proof.* Due to Lemma 1, $\widetilde{S} = \begin{bmatrix} S^t & 0 \\ 0 & 0 \end{bmatrix}$, thus $\widetilde{W}_r = \widetilde{U}\widetilde{S}\widetilde{V}^\top = USV^\top = W^t$. $\qquad\square$

We next bound the loss descent between the augmentation step and the truncation step - having performed the aggregation of the client updates.

**Theorem 7.** *Let $\widetilde{W}_r = \widetilde{U}\widetilde{S}\widetilde{V}^\top$ be the augmented factorization at global iteration $t$ and let $\widetilde{W}_r^* = \widetilde{U}\widetilde{S}^*\widetilde{V}^\top$ be the aggregated solution after client iterations, i.e., $\widetilde{S}^* = \frac{1}{C}\sum_{c=1}^{C}\widetilde{S}_c^{s_*}$. Then the variance corrected coefficient update (26) yields the guarantee*

$$
\mathcal{L}(\widetilde{W}_r^*) - \mathcal{L}(\widetilde{W}_r) \leq -(s_*\lambda)(1 - (s_*\lambda)L)\left\|\widetilde{U}^\top F(\widetilde{W}_r)\widetilde{V}\right\|^2
$$
$$
+ \left(\frac{L\lambda}{C}\sum_{c=1}^{C}\sum_{s=0}^{s_*-1}\left\|\widetilde{S}_c^s - \widetilde{S}\right\|\right)\left\|\widetilde{U}^\top F(\widetilde{W}_r)\widetilde{V}\right\| \tag{31}
$$
$$
+ \frac{L^3\lambda^2 s_*}{C}\sum_{c=1}^{C}\sum_{s=0}^{s_*-1}\left\|\widetilde{S}_c^s - \widetilde{S}_c\right\|^2.
$$

*Proof.* From (8), $P_{\widetilde{U}} = \widetilde{U}\widetilde{U}^\top$, $P_{\widetilde{V}} = \widetilde{V}\widetilde{V}^\top$, and the fact that $\widetilde{W}_{r,c}^{s=0} = \widetilde{W}_r$ for all $c = 1,\dots,C$,

$$
\widetilde{W}_{r,c}^{s_*} = \widetilde{U}\widetilde{S}_c^{s_*}\widetilde{V}^\top = \widetilde{U}\widetilde{S}_c^{s=0}\widetilde{V}^\top + \widetilde{U}\widetilde{U}^\top\sum_{s=0}^{s_*-1}\lambda\left(F_c(\widetilde{W}_{r,c}^s) - F_c(\widetilde{W}_r) + F(\widetilde{W}_r)\right)\widetilde{V}\widetilde{V}^\top
$$
$$
= \widetilde{W}_r - \lambda\sum_{s=0}^{s_*-1}P_{\widetilde{U}}F_c(\widetilde{W}_{r,c}^s)P_{\widetilde{V}} - \lambda P_{\widetilde{U}}\left(F(\widetilde{W}_r) - F_c(\widetilde{W}_r)\right)P_{\widetilde{V}}.
$$

Averaging across clients leads to

$$
\widetilde{W}_r^* = \frac{1}{C}\sum_{c=1}^{C}\widetilde{W}_{r,c}^{s_*} = \widetilde{W}_r - \frac{\lambda}{C}\sum_{c=1}^{C}\sum_{s=0}^{s_*-1}P_{\widetilde{U}}F_c(\widetilde{W}_{r,c}^s)P_{\widetilde{V}} - \frac{\lambda}{C}\sum_{c=1}^{C}P_{\widetilde{U}}\left(F(\widetilde{W}_r) - F_c(\widetilde{W}_r)\right)P_{\widetilde{V}}
$$
$$
= \widetilde{W}_r - \frac{\lambda}{C}\sum_{c=1}^{C}\sum_{s=0}^{s_*-1}P_{\widetilde{U}}F_c(\widetilde{W}_{r,c}^s)P_{\widetilde{V}}, \tag{32}
$$

where we have used the definition of the global and local gradient at $\widetilde{W}_r$, i.e., $\frac{1}{C}\sum_{c=1}^{C}F_c(\widetilde{W}_r) = F(\widetilde{W}_r)$. Based on $L$-continuity of $F$ and $F_c$, (32), and Lemma 4, we obtain further

$$
\mathcal{L}(\widetilde{W}_r^*) - \mathcal{L}(\widetilde{W}_r) \leq \left\langle\widetilde{W}_r^* - \widetilde{W}_r, F(\widetilde{W}_r)\right\rangle + \frac{L}{2}\left\|\widetilde{W}_r^* - \widetilde{W}_r\right\|^2 \tag{33}
$$
$$
= -\left\langle\frac{\lambda}{C}\sum_{c=1}^{C}\sum_{s=0}^{s_*-1}P_{\widetilde{U}}F_c(\widetilde{W}_{r,c}^s)P_{\widetilde{V}}, F(\widetilde{W}_r)\right\rangle + \frac{L}{2}\left\|\frac{\lambda}{C}\sum_{c=1}^{C}\sum_{s=0}^{s_*-1}P_{\widetilde{U}}F_c(\widetilde{W}_{r,c}^s)P_{\widetilde{V}}\right\|^2.
$$

Next, we bound each of the two right-hand-side terms separately. We first express the first term as

$$
-\left\langle \frac{\lambda}{C} \sum_{c=1}^{C} \sum_{s=0}^{s_*-1} P_{\widetilde{U}} F_c(\widetilde{W}_{r,c}^s) P_{\widetilde{V}}, F(\widetilde{W}_r) \right\rangle
$$

$$
= -\left\langle \frac{\lambda}{C} \sum_{c=1}^{C} \sum_{s=0}^{s_*-1} P_{\widetilde{U}} \left( F_c(\widetilde{W}_{r,c}^s) - F_c(\widetilde{W}_r) \right) P_{\widetilde{V}} + P_{\widetilde{U}} \left( \frac{\lambda}{C} \sum_{c=1}^{C} \sum_{s=0}^{s_*-1} F_c(\widetilde{W}_r) \right) P_{\widetilde{V}}, F(\widetilde{W}_r) \right\rangle
$$

$$
= -\left\langle \frac{\lambda}{C} \sum_{c=1}^{C} \sum_{s=0}^{s_*-1} P_{\widetilde{U}} \left( F_c(\widetilde{W}_{r,c}^s) - F_c(\widetilde{W}_r) \right) P_{\widetilde{V}} + P_{\widetilde{U}} \frac{s_*\lambda}{C} \sum_{c=1}^{C} F_c(\widetilde{W}_r) P_{\widetilde{V}}, F(\widetilde{W}_r) \right\rangle
$$

$$
= -\left\langle P_{\widetilde{U}} \left( \frac{\lambda}{C} \sum_{c=1}^{C} \sum_{s=0}^{s_*-1} F_c(\widetilde{W}_{r,c}^s) - F_c(\widetilde{W}_r) \right) P_{\widetilde{V}} + P_{\widetilde{U}} s_*\lambda F(\widetilde{W}_r) P_{\widetilde{V}}, F(\widetilde{W}_r) \right\rangle
$$

$$
= -\left\langle \widetilde{U}^\top \left( \frac{\lambda}{C} \sum_{c=1}^{C} \sum_{s=0}^{s_*-1} F_c(\widetilde{W}_{r,c}^s) - F_c(\widetilde{W}_r) \right) \widetilde{V}, \widetilde{U}^\top F(\widetilde{W}_r) \widetilde{V}^\top \right\rangle - s_*\lambda \left\langle \widetilde{U}^\top F(\widetilde{W}_r) \widetilde{V}, \widetilde{U}^\top F(\widetilde{W}_r) \widetilde{V} \right\rangle
$$

$$
= -\left\langle \frac{\lambda}{C} \sum_{c=1}^{C} \sum_{s=0}^{s_*-1} \widetilde{U}^\top \left( F_c(\widetilde{W}_{r,c}^s) - F_c(\widetilde{W}_r) \right) \widetilde{V}, \widetilde{U}^\top F(\widetilde{W}_r) \widetilde{V} \right\rangle - s_*\lambda \left\| \widetilde{U}^\top F(\widetilde{W}_r) \widetilde{V} \right\|^2,
$$

where the definitions of $P_{\widetilde{U}}$ and $P_{\widetilde{V}}$ are used. Following this, the first term then can be bounded by

$$
-\left\langle \frac{\lambda}{C} \sum_{c=1}^{C} \sum_{s=0}^{s_*-1} P_{\widetilde{U}} F_c(\widetilde{W}_{r,c}^s) P_{\widetilde{V}}, F(\widetilde{W}_r) \right\rangle
$$

$$
\leq \frac{\lambda}{C} \sum_{c=1}^{C} \sum_{s=0}^{s_*-1} \left\| \widetilde{U}^\top \left( F_c(\widetilde{W}_{r,c}^s) - F_c(\widetilde{W}_r) \right) \widetilde{V} \right\| \left\| \widetilde{U}^\top F(\widetilde{W}_r) \widetilde{V} \right\| - s_*\lambda \left\| \widetilde{U}^\top F(\widetilde{W}_r) \widetilde{V} \right\|^2
$$

$$
\leq \frac{L\lambda}{C} \sum_{c=1}^{C} \sum_{s=0}^{s_*-1} \left\| \widetilde{S}_c^s - \widetilde{S} \right\| \left\| \widetilde{U}^\top F(\widetilde{W}_r) \widetilde{V} \right\| - s_*\lambda \left\| \widetilde{U}^\top F(\widetilde{W}_r) \widetilde{V} \right\|^2,
$$

where Lemma 3 is invoked in the last inequality. Following a similar approach, we express the second term as

$$
\frac{L}{2} \left\| \frac{\lambda}{C} \sum_{c=1}^{C} \sum_{s=0}^{s_*-1} P_{\widetilde{U}} F_c(\widetilde{W}_{r,c}^s) P_{\widetilde{V}} \right\|^2 = \frac{L}{2} \left\| \frac{\lambda}{C} \sum_{c=1}^{C} \sum_{s=0}^{s_*-1} P_{\widetilde{U}} \left( F_c(\widetilde{W}_{r,c}^s) - F_c(\widetilde{W}_r) \right) P_{\widetilde{V}} + s_*\lambda P_{\widetilde{U}} F(\widetilde{W}_r) P_{\widetilde{V}} \right\|^2,
$$

which can be bounded by

$$
\frac{L}{2} \left\| \frac{\lambda}{C} \sum_{c=1}^{C} \sum_{s=0}^{s_*-1} P_{\widetilde{U}} F_c(\widetilde{W}_{r,c}^s) P_{\widetilde{V}} \right\|^2
$$

$$
\overset{(I)}{\leq} L \left\| \frac{\lambda}{C} \sum_{c=1}^{C} \sum_{s=0}^{s_*-1} P_{\widetilde{U}} \left( F_c(\widetilde{W}_{r,c}^s) - F_c(\widetilde{W}_r) \right) P_{\widetilde{V}} \right\|^2 + (s_*\lambda)^2 L \left\| P_{\widetilde{U}} F(\widetilde{W}_r) P_{\widetilde{V}} \right\|^2
$$

$$
\overset{(II)}{\leq} \frac{L}{C} \sum_{c=1}^{C} \lambda^2 s_* \sum_{s=0}^{s_*-1} \left\| P_{\widetilde{U}} \left( F_c(\widetilde{W}_{r,c}^s) - F_c(\widetilde{W}_r) \right) P_{\widetilde{V}} \right\|^2 + (s_*\lambda)^2 L \left\| P_{\widetilde{U}} F(\widetilde{W}_r) P_{\widetilde{V}} \right\|^2
$$

$$
\overset{(III)}{\leq} \frac{L^3 \lambda^2 s_*}{C} \sum_{c=1}^{C} \sum_{s=0}^{s_*-1} \left\| \widetilde{S}_c^s - \widetilde{S}_c \right\|^2 + (s_*\lambda)^2 L \left\| P_{\widetilde{U}} F(\widetilde{W}_r) P_{\widetilde{V}} \right\|^2
$$

$$
\overset{(IV)}{\leq} \frac{L^3 \lambda^2 s_*}{C} \sum_{c=1}^{C} \sum_{s=0}^{s_*-1} \left\| \widetilde{S}_c^s - \widetilde{S}_c \right\|^2 + (s_*\lambda)^2 L \left\| \widetilde{U}^\top F(\widetilde{W}_r) \widetilde{V} \right\|^2,
$$

where Lemma 5 with $\gamma = 1$ is used in in (I), Jensen's inequality is used in (II), Lemma 3 is used in in (III), and (IV) follows from the Operator norm inequality of the Frobenius norm in combination with orthonormality of $U$ and $V^\top$.

Plugging these two bounds into (33) gives

$$\mathcal{L}(\widetilde{W}_r^*) - \mathcal{L}(\widetilde{W}_r) \le - \left\langle \frac{\lambda}{C} \sum_{c=1}^{C} \sum_{s=0}^{s_*-1} P_{\widetilde{U}} F_c(\widetilde{W}_{r,c}^s) P_{\widetilde{V}}, F(\widetilde{W}_r) \right\rangle + \frac{L}{2} \left\| \frac{\lambda}{C} \sum_{c=1}^{C} \sum_{s=0}^{s_*-1} P_{\widetilde{U}} F_c(\widetilde{W}_{r,c}^s) P_{\widetilde{V}} \right\|^2$$

$$\le \frac{L\lambda}{C} \sum_{c=1}^{C} \sum_{s=0}^{s_*-1} \left\| \widetilde{S}_c^s - \widetilde{S} \right\| \left\| \widetilde{U}^\top F(\widetilde{W}_r) \widetilde{V} \right\| - s_* \lambda \left\| \widetilde{U}^\top F(\widetilde{W}_r) \widetilde{V} \right\|^2$$

$$+ \frac{L^3 \lambda^2 s_*}{C} \sum_{c=1}^{C} \sum_{s=0}^{s_*-1} \left\| \widetilde{S}_c^s - \widetilde{S}_c \right\|^2 + (s_*\lambda)^2 L \left\| \widetilde{U}^\top F(\widetilde{W}_r) \widetilde{V} \right\|^2$$

$$= - (s_*\lambda)(1 - (s_*\lambda)L) \left\| \widetilde{U}^\top F(\widetilde{W}_r) \widetilde{V} \right\|^2$$

$$+ \left( \frac{L\lambda}{C} \sum_{c=1}^{C} \sum_{s=0}^{s_*-1} \left\| \widetilde{S}_c^s - \widetilde{S} \right\| \right) \left\| \widetilde{U}^\top F(\widetilde{W}_r) \widetilde{V} \right\|$$

$$+ \frac{L^3 \lambda^2 s_*}{C} \sum_{c=1}^{C} \sum_{s=0}^{s_*-1} \left\| \widetilde{S}_c^s - \widetilde{S}_c \right\|^2,$$

which concludes the proof. $\qquad \square$

With this result, we next bound the loss descent between the augmentation and coefficient aggregation step in the following theorem.

**Theorem 8.** *Under the same assumptions as in Theorem 7. Let the local learning rate be $0 < \lambda \le \frac{1}{12Ls_*}$ with number of local iterations $s_* \ge 1$. Then,*

$$\mathcal{L}(\widetilde{W}_r^*) - \mathcal{L}(\widetilde{W}_r) \le -s_*\lambda(1 - 12s_*\lambda L) \left\| \widetilde{U}^\top F(\widetilde{W}_r) \widetilde{V} \right\|^2. \tag{34}$$

*Proof.* Applying the drift bound given in Theorem 1 to the loss descent bound given by Theorem 7 in (31) leads to

$$- (s_*\lambda)(1 - (s_*\lambda)L) \left\| \widetilde{U}^\top F(\widetilde{W}_r) \widetilde{V} \right\|^2$$

$$+ \left( \frac{L\lambda}{C} \sum_{c=1}^{C} \sum_{s=0}^{s_*-1} \left( \exp(1) s_* \lambda \left\| \widetilde{U}^\top F(\widetilde{W}_r) \widetilde{V} \right\| \right) \right) \left\| \widetilde{U}^\top F(\widetilde{W}_r) \widetilde{V} \right\|$$

$$+ \frac{L^3 \lambda^2 s_*}{C} \sum_{c=1}^{C} \sum_{s=0}^{s_*-1} \left( \exp(1) s_* \lambda \left\| \widetilde{U}^\top F(\widetilde{W}_r) \widetilde{V} \right\| \right)^2$$

$$= - (s_*\lambda)(1 - (s_*\lambda)L) \left\| \widetilde{U}^\top F(\widetilde{W}_r) \widetilde{V} \right\|^2 + L\lambda^2 s_*^2 \exp(1) \left\| \widetilde{U}^\top F(\widetilde{W}_r) \widetilde{V} \right\|^2$$

$$+ L^3 \lambda^4 s_*^4 \exp(2) \left\| \widetilde{U}^\top F(\widetilde{W}_r) \widetilde{V} \right\|^2$$

$$= - (s_*\lambda)(1 - (s_*\lambda)L - (s_*\lambda)L\exp(1) - (s_*\lambda)^3 L^2 \exp(2)) \left\| \widetilde{U}^\top F(\widetilde{W}_r) \widetilde{V} \right\|^2$$

$$\le - (s_*\lambda)(1 - (s_*\lambda)L(1 + \exp(1) + \exp(2))) \left\| \widetilde{U}^\top F(\widetilde{W}_r) \widetilde{V} \right\|^2$$

$$\le - (s_*\lambda)(1 - 12(s_*\lambda)L) \left\| \widetilde{U}^\top F(\widetilde{W}_r) \widetilde{V} \right\|^2,$$

where we have used that $(s_*\lambda)L \le 1$ and that $1 + \exp(1) + \exp(2) \approx 11.107 \le 12$. $\qquad \square$

We are now prepared to prove Theorem 2, which we restate in terms of Riemannian gradients as below.

**Theorem 9.** *(Restatement of Theorem 2) Let $U^t S^t V^{t,\top}$ and $U^{t+1} S^{t+1} V^{t+1,\top}$ be the factorization before and after iteration $t$ of Algorithm 1 with variance correction and singular value truncation threshold $\vartheta$. Let $\mathcal{L}_c$ and $\mathcal{L}$ be $L$-smooth with constant $L$, and let the local learning rate be $0 \leq \lambda \leq \frac{1}{12 L s_*}$. Then the global loss descent is bounded by*

$$\mathcal{L}(U^{t+1} S^{t+1} V^{t+1,\top}) - \mathcal{L}(U^t S^t V^{t,\top}) \leq -(s_*\lambda)(1 - 12(s_*\lambda)L)\left\|\widetilde{U}^\top F(\widetilde{W}_r)\widetilde{V}\right\|^2 + L\vartheta. \quad (35)$$

*Proof.* Consider $\mathcal{L}(W_r^{t+1})$ and $\mathcal{L}(\widetilde{W}_r^*)$, i.e., the loss values before and after the truncation step. By the mean value theorem, we obtain for some $h \in [0, 1]$

$$\begin{aligned}
\mathcal{L}(W_r^{t+1}) &= \mathcal{L}(\widetilde{W}_r^*) + \left\langle -F(hW_r^{t+1} + (1-h)\widetilde{W}_r^*), W_r^{t+1} - \widetilde{W}_r^* \right\rangle \\
&\leq \mathcal{L}(\widetilde{W}_r^*) + \left\|F(hW_r^{t+1} + (1-h)\widetilde{W}_r^*)\right\| \left\|W_r^{t+1} - \widetilde{W}_r^*\right\| \quad (36) \\
&\leq \mathcal{L}(\widetilde{W}_r^*) + L\vartheta
\end{aligned}$$

where $L$-smoothness and the fact that $\vartheta \geq \left\|W_r^{t+1} - \widetilde{W}_r^*\right\|$ are used in (II), where the latter follows from the singular value truncation threshold. Combining the above arguments with Lemma 7 and Theorem 8 yields

$$\begin{aligned}
\mathcal{L}(W_r^{t+1}) - \mathcal{L}(W_r^t) &= (\mathcal{L}(W_r^{t+1}) - \mathcal{L}(\widetilde{W}_r^*)) + (\mathcal{L}(\widetilde{W}_r^*) - \mathcal{L}(\widetilde{W}_r)) + (\mathcal{L}(\widetilde{W}_r) - \mathcal{L}(W_r^t)) \\
&\leq L\vartheta - (s_*\lambda)(1 - 12(s_*\lambda)L)\left\|\widetilde{U}^\top F(\widetilde{W}_r)\widetilde{V}\right\|^2,
\end{aligned}$$

which concludes the proof. $\qquad\square$

### H.3 GLOBAL CONVERGENCE OF FEDLRT WITH FULL VARIANCE CORRECTION

**Theorem 10.** *(Restatement of Theorem 3) Assume that $\mathcal{L}$ is $L$-smooth with constant $L$ for all $c = 1, \ldots, C$. Let $\widetilde{U}^t \widetilde{S}^t \widetilde{V}^{t,\top}$ be the augmented representation at iteration $t$. Then Algorithm 1 guarantees for the learning rate $\lambda \leq \frac{1}{12 L s_*}$ and final iteration $T$*

$$\min_{t=1,\ldots,T} \left\|\nabla_{\widetilde{S}} \mathcal{L}(U^t S^t V^{t,\top})\right\|^2 \leq \frac{48L}{T}\left(\mathcal{L}(W_r^{t=1}) - \mathcal{L}(W_r^{t=T+1})\right) + 48L^2\vartheta. \quad (37)$$

*Proof.* Consider Theorem 2,

$$\mathcal{L}(W_r^{t+1}) - \mathcal{L}(W_r^t) \leq L\vartheta - (s_*\lambda)(1 - 12(s_*\lambda)L)\left\|\nabla_{\widetilde{S}} \mathcal{L}(U^t S^t V^{t,\top})\right\|^2, \quad (38)$$

and assume that $\lambda s_* = \frac{1}{24L}$, i.e. $\lambda = \frac{1}{24 L s_*} \leq \frac{1}{L s_*}$, which obeys the learning rate requirement of Theorem 2. Plugging this learning rate into (38) gives

$$\left\|\nabla_{\widetilde{S}} \mathcal{L}(U^t S^t V^{t,\top})\right\|^2 \leq 48L\left(\mathcal{L}(W_r^t) - \mathcal{L}(W_r^{t+1}) + L\vartheta\right).$$

Averaging from $t = 1$ to $t = T$ yields

$$\begin{aligned}
\min_{t=1,\ldots,T}\left\|\nabla_{\widetilde{S}}\mathcal{L}(U^t S^t V^{t,\top})\right\|^2 &\leq \frac{1}{T}\sum_{t=1}^{T}\left\|\nabla_{\widetilde{S}}\mathcal{L}(U^t S^t V^{t,\top})\right\|^2 \\
&\leq \frac{48L}{T}\left(\mathcal{L}(W_r^{t=1}) - \mathcal{L}(W_r^{t=T+1})\right) + 48L^2\vartheta,
\end{aligned}$$

which concludes the proof. $\qquad\square$

We remark that for a general loss function, it is possible that a point with small gradient magnitude can be far from the stationary points. However, assuming that the loss function is locally strongly convex in a neighborhood of a stationary point, then the gradient magnitude can be used to bound the distance to this stationary point in the neighborhood. For further reference, we point to (**?**, Eq. (4.12)) for the estimate.

## I    ANALYSIS FOR FEDLRT WITH SIMPLIFIED VARIANCE CORRECTION

We consider the FeDLRT method with simplified variance correction, see Algorithm 5. Key difference to the standard FeDLRT with full variance correction, see Algorithm 1 is the modified coefficient update, incorporating global gradient information of the non-augmented coefficient matrix $S$ for the variance correction term, that is

$$\check{V}_c = \check{G}_{\widetilde{S}} - \check{G}_{\widetilde{S},c} = \begin{bmatrix} \nabla_S \mathcal{L}(U^t S^t V^{t,\top}) - \nabla_S \mathcal{L}_c(U^t S^t V^{t,\top}) & 0 \\ 0 & 0 \end{bmatrix}. \tag{39}$$

Using the Riemmanian gradient, we can equivalently write

$$\check{V}_c = \begin{bmatrix} U^\top | 0 \end{bmatrix} (F(\widetilde{W}_r) - F_c(\widetilde{W}_r)) \begin{bmatrix} V \\ 0 \end{bmatrix} = \widetilde{U}^\top \begin{bmatrix} I & 0 \\ 0 & 0 \end{bmatrix} (F_c(\widetilde{W}_r) - F(\widetilde{W}_r)) \begin{bmatrix} I & 0 \\ 0 & 0 \end{bmatrix} \widetilde{V}.$$

Remember the simplified variance corrected local coefficient update, given by

$$\widetilde{S}_c^{s+1} = \widetilde{S}_c^s + \lambda \widetilde{U}^\top \left( F_c(\widetilde{W}_{r,c}^s) + \begin{bmatrix} I & 0 \\ 0 & 0 \end{bmatrix} (F_C(\widetilde{W}_r) - F(\widetilde{W}_r)) \begin{bmatrix} I & 0 \\ 0 & 0 \end{bmatrix} \right) \widetilde{V}$$

$$= \widetilde{S}_c^s + \lambda \widetilde{U}^\top \left( F_c(\widetilde{W}_{r,c}^s) \right) \widetilde{V} + \check{V}_c. \tag{40}$$

### I.1    GLOBAL LOSS DESCENT FOR FEDLRT WITH SIMPLIFIED VARIANCE CORRECTION

In the following we provide proof for a global loss descent for Algorithm 5, i.e. using the local coefficient update with variance correction (40).

**Theorem 11.** *(Restatement of Theorem 4) Under Assumption 1, if the local learning rate $0 < \lambda \leq \frac{1}{12Ls_*}$, then Algorithm 5 leads to the global loss descent*

$$\mathcal{L}(W_r^{t+1}) - \mathcal{L}(W_r^t) \leq -s_* \lambda (1 - \delta^2 - 12s_* \lambda L + \delta^2 s_* \lambda) \left\| \widetilde{U}^\top F(\widetilde{W}_r) \widetilde{V} \right\|^2 + L\vartheta, \tag{41}$$

*with $W_r^t = U^t S^t V^{t,\top}$ and $W_r^{t+1} = U^{t+1} S^{t+1} V^{t+1,\top}$.*

*Proof.* We split the adjusted coefficient update in (40) into the non-augmented $r \times r$ matrix $S$ and the tree off-diagonal blocks given by the augmentation $\widehat{S}$:

$$\widehat{S} = \widetilde{S} - \begin{bmatrix} S & 0 \\ 0 & 0 \end{bmatrix}. \tag{42}$$

Analogously to the proof of Theorem 2, we consider

$$\mathcal{L}(\widetilde{W}_r^*) - \mathcal{L}(\widetilde{W}_r) \leq \left\langle \widetilde{W}_r^* - \widetilde{W}_r, F(\widetilde{W}_r) \right\rangle + \frac{L}{2} \left\| \widetilde{W}_r^* - \widetilde{W}_r \right\|^2$$

$$= \left\langle \widetilde{U}\widetilde{S}^*\widetilde{V}^\top - \widetilde{U}\widetilde{S}\widetilde{V}^\top, F(\widetilde{W}_r) \right\rangle + \frac{L}{2} \left\| \widetilde{U}\widetilde{S}^*\widetilde{V}^\top - \widetilde{U}\widetilde{S}\widetilde{V}^\top \right\|^2$$

$$= \left\langle \widetilde{S}^* - \widetilde{S}, \widetilde{U}^\top F(\widetilde{W}_r)\widetilde{V} \right\rangle + \frac{L}{2} \left\| \widetilde{S}^* - \widetilde{S} \right\|^2$$

$$= \left\langle \widetilde{S}^* - \widetilde{S}, -\nabla_{\widetilde{S}} \mathcal{L}(\widetilde{W}_r) \right\rangle + \frac{L}{2} \left\| \widetilde{S}^* - \widetilde{S} \right\|^2,$$

where the transformation uses orthonormality of $\widetilde{U}$ and $\widetilde{V}$ and definition of the projected gradient. We split the right hand side in terms corresponding to augmented terms $\widehat{S}$ and non-augmented terms $S$ according to (42), i.e.,

$$\left\langle S^* - S, -\nabla_S \mathcal{L}(\widetilde{W}_r) \right\rangle + \frac{L}{2} \left\| S^* - S \right\|^2, \tag{43}$$

which is treated exactly as in the proof of Theorem 2, and the augmented terms

$$\left\langle \widehat{S}^* - \widehat{S}, -\nabla_{\widehat{S}} \mathcal{L}(\widetilde{W}_r) \right\rangle + \frac{L}{2} \left\| \widehat{S}^* - \widehat{S} \right\|^2. \tag{44}$$

First we bound the term (43). Remember that $\widehat{S} = 0$ at the start of the local iterations due to orthonormality of $\widetilde{U}, \widetilde{V}$. The coefficient update (40) for $S$ reads

$$S_c^{s+1} = S_c^s + \lambda U^\top \left( F_c(\widetilde{W}_{r,c}^s) - F_c(\widetilde{W}_r) + F(\widetilde{W}_r) \right) V. \tag{45}$$

Then we can readily apply Theorem 2 to obtain the bound

$$\left\langle S^* - S, -\nabla_S \mathcal{L}(\widetilde{W}_r) \right\rangle + \frac{L}{2} \|S^* - S\|^2 \leq -(s_* \lambda)(1 - 12(s_* \lambda)L) \left\| U^\top F(\widetilde{W}_r)V \right\|^2. \tag{46}$$

Next, we bound (44), starting with the first term:

$$\left\langle \widehat{S}^* - \widehat{S}, -\nabla_{\widehat{S}} \mathcal{L}(\widetilde{W}_r) \right\rangle \overset{(I)}{=} \left\langle \widehat{S}^* - 0, -\nabla_{\widehat{S}} \mathcal{L}(\widetilde{W}_r) \right\rangle$$

$$= \left\langle -\frac{\lambda}{C} \sum_{c=1}^{C} \sum_{s=0}^{s_*-1} \nabla_{\widehat{S}} \mathcal{L}_c(\widetilde{W}_{r,c}^s), -\nabla_{\widehat{S}} \mathcal{L}(\widetilde{W}_r) \right\rangle$$

$$= \frac{\lambda}{C} \sum_{c=1}^{C} \sum_{s=0}^{s_*-1} \left\langle \nabla_{\widehat{S}} \mathcal{L}_c(\widetilde{W}_{r,c}^s), \nabla_{\widehat{S}} \mathcal{L}(\widetilde{W}_r) \right\rangle$$

$$\leq \frac{\lambda}{C} \sum_{c=1}^{C} \sum_{s=0}^{s_*-1} \left\| \nabla_{\widehat{S}} \mathcal{L}_c(\widetilde{W}_{r,c}^s) \right\| \left\| \nabla_{\widehat{S}} \mathcal{L}(\widetilde{W}_r) \right\|$$

$$\overset{(II)}{\leq} \frac{\lambda}{C} \sum_{c=1}^{C} \sum_{s=0}^{s_*-1} \delta^2 \left\| \nabla_{\widetilde{S}} \mathcal{L}(\widetilde{W}_r) \right\| \left\| \nabla_{\widetilde{S}} \mathcal{L}(\widetilde{W}_r) \right\|$$

$$= \delta^2 s_* \lambda \left\| \nabla_{\widetilde{S}} \mathcal{L}(\widetilde{W}_r) \right\|^2 = \delta^2 s_* \lambda \left\| \widetilde{U}^\top F(\widetilde{W}_r)\widetilde{V} \right\|^2,$$

where we use $\widehat{S} = 0$ in (I), and Assumption 1 in (II). Next, we bound the second term

$$\frac{L}{2} \left\| \widehat{S}^* - \widehat{S} \right\|^2 = \frac{L}{2} \left\| -\frac{\lambda}{C} \sum_{c=1}^{C} \sum_{s=0}^{s_*-1} \nabla_{\widehat{S}} \mathcal{L}(\widetilde{W}_{r,c}^S) \right\|^2$$

$$\overset{(I)}{\leq} \frac{L}{2} \lambda^2 \frac{1}{C} \sum_{c=1}^{C} \left\| \sum_{s=0}^{s_*-1} \nabla_{\widehat{S}} \mathcal{L}(\widetilde{W}_{r,c}^S) \right\|^2$$

$$\overset{(I)}{\leq} \frac{L}{2} s_* \lambda^2 \frac{1}{C} \sum_{c=1}^{C} \sum_{s=0}^{s_*-1} \left\| \nabla_{\widehat{S}} \mathcal{L}(\widetilde{W}_{r,c}^S) \right\|^2$$

$$\leq s_* \frac{L}{2} \delta^2 \lambda^2 \frac{1}{C} \sum_{c=1}^{C} \sum_{s=0}^{s_*-1} \left\| \nabla_{\widetilde{S}} \mathcal{L}(\widetilde{W}_r) \right\|^2$$

$$\leq \frac{L}{2} \delta^2 (s_* \lambda)^2 \left\| \nabla_{\widetilde{S}} \mathcal{L}(\widetilde{W}_r) \right\|^2 = \frac{L}{2} \delta^2 (s_* \lambda)^2 \left\| \widetilde{U}^\top F(\widetilde{W}_r)\widetilde{V} \right\|^2,$$

where we used Jensen's inequality in (I) again Assumption 1. We combine the bound on the non-augmented terms (46) and the two bounds above for the augmented terms to

$$\mathcal{L}(\widetilde{W}_r^*) - \mathcal{L}(\widetilde{W}_r) \leq \left\langle \widetilde{W}_r^* - \widetilde{W}_r, F(\widetilde{W}_r) \right\rangle + \frac{L}{2} \left\| \widetilde{W}_r^* - \widetilde{W}_r \right\|^2$$

$$\leq -(s_* \lambda)(1 - 12(s_* \lambda)L) \left\| U^\top F(\widetilde{W}_r)V \right\|^2 + \delta s_* \lambda \left\| \widetilde{U}^\top F(\widetilde{W}_r)\widetilde{V} \right\|^2 + \delta(s_* \lambda)^2 \left\| \widetilde{U}^\top F(\widetilde{W}_r)\widetilde{V} \right\|^2$$

$$\overset{(I)}{\leq} -(s_* \lambda)(1 - 12(s_* \lambda)L) \left\| \widetilde{U}^\top F(\widetilde{W}_r)\widetilde{V} \right\|^2 + \delta s_* \lambda \left\| \widetilde{U}^\top F(\widetilde{W}_r)\widetilde{V} \right\|^2 + \delta(s_* \lambda)^2 \left\| \widetilde{U}^\top F(\widetilde{W}_r)\widetilde{V} \right\|^2$$

$$= -(s_* \lambda)(1 - \delta^2 - 12(s_* \lambda)L + \delta^2(s_* \lambda)) \left\| \widetilde{U}^\top F(\widetilde{W}_r)\widetilde{V} \right\|^2,$$

where we use in (I) $\left\| U^\top F(\widetilde{W}_r)V \right\| \leq \left\| \widetilde{U}^\top F(\widetilde{W}_r)\widetilde{V} \right\|$. Using Equation (36), we can conclude the proof:

$$\mathcal{L}(U^{t+1}S^{t+1}V^{t+1,\top}) - \mathcal{L}(U^t S^t V^{t,\top})$$

$$\leq -(s_*\lambda)(1 - \delta^2 - 12(s_*\lambda)L + \delta^2(s_*\lambda)) \left\| \widetilde{U}^\top F(\widetilde{W}_r)\widetilde{V} \right\|^2 + L\vartheta.$$

$\square$

### I.2 GLOBAL CONVERGENCE OF FEDLRT WITH SIMPLIFIED VARIANCE CORRECTION

**Corollary 2.** *(Restatement of Corollary 1) Under Assumption 1, Algorithm 5 guarantees for the learning rate $\lambda \leq \frac{1}{s_*(12L+\delta^2)}$*

$$\min_{t=1,\ldots,T} \left\| \nabla_{\widetilde{S}}\mathcal{L}(W_r{}^t) \right\|^2 \leq \frac{96L}{T} \left( \mathcal{L}(W_r{}^1) - \mathcal{L}(W_r{}^{T+1}) \right) + 96L^2\vartheta, \tag{47}$$

*with $W_r{}^t = U^t S^t V^{t,\top}$, $W_r{}^1 = U^1 S^1 V^{1,\top}$. and $W_r{}^{T+1} = U^{T+1}S^{T+1}V^{T+1,\top}$.*

*Proof.* Consider Theorem 4,

$$\mathcal{L}(W_r{}^{t+1}) - \mathcal{L}(W_r{}^t) \leq -(s_*\lambda)(1 - \delta^2 - 12(s_*\lambda)L + \delta^2(s_*\lambda)) \left\| \widetilde{U}^\top F(\widetilde{W}_r)\widetilde{V} \right\|^2 + L\vartheta$$

and assume that $\lambda s_* = \frac{1}{(12L+\delta^2)}$, i.e. $\lambda = \frac{1}{s_*(12L+\delta^2)} \leq \frac{1}{Ls_*}$, which obeys the learning rate requirement of Theorem 2. Plugging this learning rate into (38) gives

$$\left\| \nabla_{\widetilde{S}}\mathcal{L}(W_r{}^t) \right\|^2 \leq 96L \left( \mathcal{L}(W_r{}^t) - \mathcal{L}(W_r{}^{t+1}) + L\vartheta \right),$$

where we use $\left(\frac{1}{4} - \delta^2\right) \leq \frac{1}{4}$ and $\frac{1}{(12L+\delta^2)} \leq \frac{1}{12L}$ Averaging from $t = 1$ to $t = T$ yields

$$\min_{t=1,\ldots,T} \left\| \nabla_{\widetilde{S}}\mathcal{L}(W_r{}^t) \right\|^2 \leq \frac{1}{T} \sum_{t=1}^{T} \left\| \widetilde{U}^\top F(\widetilde{W}_r)\widetilde{V} \right\|^2$$

$$\leq \frac{96L}{T} \left( \mathcal{L}(W_r{}^{t=1}) - \mathcal{L}(W_r{}^{t=T+1}) \right) + 96L^2\vartheta,$$

which concludes the proof. $\square$

