# OpenReview forum: "Federated Dynamical Low-Rank Training with Global Loss Convergence Guarantees"
_ICLR.cc/2025/Conference — Submitted to ICLR 2025_

### Official Review · Reviewer_c2me · 2024-10-28

**Soundness:** 2
**Presentation:** 2
**Contribution:** 3
**Rating:** 5
**Confidence:** 3

**Summary:**

To reduce client computing and communication costs, this paper proposes a dynamic low-rank training method. The algorithm creates a global low-rank basis of network weights, enabling client training on a small coefficient matrix. This paper also gives the theoretical convergence guarantee of the algorithm with the variance correction scheme.The authors provide empirical validations on the theoretical results as well.

**Strengths:**

1. The proposed method is well-motivated, the paper reduces the number of training parameters using the dynamic low-rank approximation method with less performance sacrifice.
2. This paper gives the convergence analysis of the federated dynamical low-rank training (FeDLRT) method.
3. The empirical results show that the federated dynamical low-rank training (FeDLRT) method is better than FedLin.

**Weaknesses:**

1. The method only considers the full participation case without partial participation.
2. In addition to FedLin, other low-rank training methods should be considered such as [1].
3. In experiments, the number of clients is too small. In more realistic settings, more clients and partial participation should be considered.

[1] Nam Hyeon-Woo, Moon Ye-Bin, Tae-Hyun Oh. FedPara: Low-rank Hadamard Product for Communication-Efficient Federated Learning. ICLR 2022.

**Questions:**

See in weaknesses.

---

> ### Author Response · Authors · 2024-11-25
>
> We thank the reviewer for their thoughtfull review and appreciate their feedback.
> We are happy that they found our work well motivated and that they liked the convergence analysis supported by empirical
> test cases.
>
> Below we answer to the reviewers feedback
>
> ### W1. and W3. **Partial client participation**:
>
> Thank you for the suggestion. The proposed FeDLRT is indeed able to accommodate partial client participation. We have
> added a test case with $200$ clients and partial participation using $10$ randomly sampled active clients for training
> Resnet18 on CIFAR10 in the revised manuscript in Figure 5. The result indicates that FeDLRT matches the performance of
> the FedLin method with reduced communication, computation, and memory cost, as observed in the full participation case.
>
> ---
>
> ### W2 **Comparison to other approaches**
>
> Thank you for suggesting the comparison. We expand the introduction section to include a more detailed discussion about
> existing FL methods with low-rank and sparse compression and their comparison to FeDLRT.
>
> The primary difference between FeDLRT and many other low-rank or sparse compression methods, e.g., FeDLR, Comfetch, and
> FetchSGD, is that FeDLRT only trains the low rank coefficient matrix on the clients, which leads to much lower memory
> and compute costs on clients. On the other hand, by introducing the global basis, FeDLRT avoids the assembly of the full
> rank weight matrices on the server and leads to convergence guarantees, which are not available in existing methods that
> train only the low rank or sparse components (sub-models) on the clients, such as FedHM, FedPara, FjORD, and PriSM.
> The detailed comparison between FeDLRT and other methods in terms of compute, memory, and communication costs on both
> the server and client is now given in Table 1 in the manuscript.

---

### Official Review · Reviewer_HJWB · 2024-11-03

**Soundness:** 3
**Presentation:** 3
**Contribution:** 2
**Rating:** 5
**Confidence:** 3

**Summary:**

The paper considers federated learning training with dynamical low-rank training scheme that allows to reduce both computations, since the training is performed directly in the low-rank space, as well as communications as only low-rank representations of neural networks need to be communicated instead of the full layers. This allows for efficient training in the low-resource & low-communication FL setup.

**Strengths:**

The paper is well written and easy to follow.
The paper provides proof of convergence of their proposed method when local gradients on clients are deterministic.
The paper provides a way of combining their algorithm with variance correction due to difference in data distribution on different clients.
The paper provides experimental verification of their proposed method.

**Weaknesses:**

1. The paper provides convergence guarantees for their method only if the gradients are deterministic, which is limiting the potential applications of their algorithm.

Experimental evaluation is limited:

2. Experimental results are provided only for the artificial dataset, and for cifar 10 and cifar 100 datasets, which are not FL-related benchmarks. Evaluating on more realistic FL datasets, such as [R1, R2] would make the paper stronger.

3. Paper does not give all of the details of experiments, thus not reproducible. Several key details that are missing:
3.1. How learning rate was chosen in Cifar-10 experiments
3.2. How the data was split across the clients in Cifar-10 experiments.
3.3. On Cifar-10 experiments first warm-up is performed on 1 client, but how that 1 client was chosen is not mentioned in the paper.

4. Paper does not tune the learning rate, but uses the same learning rate for different algorithms - that might give advantage to some algorithms. Tuning the learning rate is needed for fair comparison.

5. Paper shows only 93% of accuracy of Cifar10 for 32 workers (I believe it is coming from suboptimal stepsize schedule that is just constant over training), which is quite low and quite far from stoa Cifar10 accuracy. Analysis of how close the proposed algorithm can reach to stoa accuracy is missing.


[R1] Terrail et al, FLamby: Datasets and Benchmarks for Cross-Silo Federated Learning in Realistic Healthcare Settings.
[R2] Caldas et al, LEAF: A Benchmark for Federated Settings.

**Questions:**

1. In table 1, what is “b”? I think “b” was not introduced before. Is it a batchsize?
2. Why in Figure 6, when clients increase from 1 to 4 , accuracy increases, but later decreases? This is not intuitive to me.
3. It is also not very intuitive to me - why does the accuracy drop with a larger number of clients? Since with larger number clients, the number of local steps is smaller, it might point to the better performance of methods without variance reduction, such as FedAvg. I am also wondering if such behavior comes from suboptimal choice of hyperparameters.
4. Could you also plot confidence intervals together with the median?
5. Table 1 is slightly too small, if printed on the paper.
6. I did not fully understand the algorithm. Why do you need to do basis augmentation and what does it mean? Also why is the $\tilde S^\star$ not diagonal?

---

> ### Author Response · Authors · 2024-11-25
>
> We thank the reviewer for their constructive feedback. We appreciate the assessment that the paper is well written and
> easy to follow.
> We answer to the reviewers feedback below
>
> ### W1 Convergence Guarantees
>
> We would like to clarify that, even though the convergence analysis in this paper considers only the case when the
> gradients are deterministic, FeDLRT could converge when stochastic gradients are used. In fact, stochastic gradients are
> used in all neural network training results reported in this paper, which empirically verifies the feasibility of using
> FeDLRT with stochastic gradients. Extending the convergence analysis of FeDLRT to the case of stochastic gradients is in
> the scope of future work. We do not foresee obstacles on the low rank training side and expect the challenge to come
> from the variance correction scheme, which could potentially be resolved by adopting variance correction schemes that
> are tailored to stochastic gradients.
>
> ---
>
> ### W2 Datasets
>
> Thank you for the suggestion. We agree that adding tests on FL specific benchmarks would strengthen the paper. However,
> since most of the publications on compression and communication efficient FL methods use CIFAR-10 and CIFAR-100 as test
> problems, we prioritize these more generic/artificial datasets as the benchmarks considered in this paper.
>
> ---
>
> ### W3 Experimental Details
>
> The hyperparameters for the performed experiments can be found in Table 2 in Appendix C. The data in the original
> numerical results was distributed homogeneously across clients. We have added new numerical results for a) partial
> participation b) non i.i.d. data distribution and c) to evaluate effective communication cost in Section 4.2. of the
> revised manuscript.
>
> ---
>
> ### W4 Learning Rate
>
> Thank you for this question. A key advantage of our method is that the low-rank optimization trajectory obtained by
> FeDLRT is close to the trajectory of the corresponding full-rank method, as proven in Theorem 5 in the appendix.
> Notably, a consequence of this theorem is that the step-size for the low-rank scheme is not dependent on the local
> geometry and curvature of the low-rank manifold on which it operates. In particular, this means that the low-rank scheme
> can use the same learning rates as the corresponding full-rank scheme (as FedLin or FedAvg).
>
> Thus, a straightforward approach to determine the learning rate is to tune it for the full-rank scheme, e.g. FedLin and
> FedAvg, and simply apply it to FeDLRT. This is exactly how we obtained the learning rates in this work.
>
>
> ---
>
> ### W5 SOTA Accuracy
>
> Certainly, our methods are extendable to learning rate schedulers, as long as the learning rates obey the step size
> restrictions of Theorems 2, 3 and 4, respectively. For the scope of this work, we refrained from further tuning the
> learning rate via schedulers to keep the experimental setup as simple as possible.
> However, comparison with, e.g. [1] shows that approximately 93\% accuracy for $32$ workers on CIFAR10-Resnet18 is a
> reasonable result. In [1] the authors report in the same dataset with 10 local updates and $20$ clients at approximately
> $93.43\%$ accuracy.
>
> We remark that our method provides strong analysis that shows how close to SOTA one can get. Theorem 5 in the appendix
> describes a rigorous error bound for the difference between our low-rank method FeDLRT and a full-rank scheme as FedLin.
> This means, FeDLRT is able to mirror the performance of FedLIN with the same learning hyperparameters up to truncation
> tolerance. Our experimental results verify this result empirically.
>
> [1] FedHM: Efficient Federated Learning for Heterogeneous Models via Low-rank Factorization
> Dezhong Yao, Wanning Pan, Michael J O'Neill, Yutong Dai, Yao Wan, Hai Jin, Lichao Sun
>
> ---
>
> ### Q1 batch size b
>
> Thank you for catching that. Indeed, $b$ denotes the batch size. We have added this to the description of Table 1.
>
> ---
>
> ### Q2 and Q3 Accuracy changes with respect to the client number
>
> Thank you for this question. In Figure 7 (new manuscript), the validation accuracy improves slightly by
> approximately 0.25% when the number of clients increases from 1 to 4 and then decreases as the client number increases. We attribute this effect to the variance reduction scheme, which mitigates the client drift as shown in Figure 7. However, we do not have specific explanations on the slight accuracy increase from 1 to 4 clients. One potential cause of this effect is that we train the networks by using stochastic gradient descent with a variance correction scheme that was designed for full (deterministic) gradients. We plan to extend the implementation and analysis of FeDLRT to incorporate other variance correction schemes for stochastic gradients in the future work.
>
> ---
>
> ### Q4 Confidence intervalls
>
> In Figure 6 (now Figure 7 in the appendix of the revised manuscript), we use a boxplot to represent the statistics of
> our results. The box denotes the 25-75 percentile interval.

---

> > ### Author Response · Authors · 2024-11-25
> >
> > ### Q5 Table 1 too small
> >
> > In the revised manuscript, we simplified the $\mathcal{O}$-expression in table 1 by keeping only the leading order terms
> > and increased the font size for better readability.
> >
> > ---
> >
> > ### Q6 Explanations for the Algorithm
> >
> > We have designed the FeDLRT method to adaptively compress the neural network during federated learning. Since the ideal
> > rank of the low-rank representation is unknown a-priori, we developed the basis augmentation mechanic to allow for
> > adjustments of the rank during training. In short, basis augmentation (line 5 in Algorithm 1) extends the old basis
> > representation of the network weights by adding new bases, which are given by the gradient dynamics of the network, and
> > thus doubles the rank from $r$ to $2r$.
> > On the other hand, rank truncation (line 17 in Algorithm 1) is performed after the aggregation step by applying a
> > truncated SVD to the augmented, aggregated coefficient matrix $\widetilde{S}^*$ of size $2r\times2r$, which reduces the
> > rank from $2r$ to the minimal rank satisfying the truncation tolerance.
> >
> > $\widetilde{S}$ is diagonal after the truncation step (lines 17 and 18 in Algorithm 1). However, during the local
> > training (line 13 or 15 in Algorithm~1), all entries of the local coefficient matrix $\widetilde{S}_c$ can become
> > nonzero via the gradient descent updates. Therefore, the coefficients may not be diagonal after local steps.

---

### Official Review · Reviewer_P7tt · 2024-11-03

**Soundness:** 3
**Presentation:** 2
**Contribution:** 3
**Rating:** 8
**Confidence:** 2

**Summary:**

In this work, the authors propose a new algorithm to perform the task of horizontal federated learning. In this particular setting, all clients share the same model architecture and data features. The compute available on the client side is fairly limited and the costs associated with the communication between the central server and the clients are expensive. Current state of the art federated learning algorithms suffer from memory, compute and communication costs that scale quadratically with the size of the problem.

To address the aforementioned issues, the authors propose a federated dynamical low-rank training scheme (FeDLRT) with and without variance correction, incorporating both a full and a simple variance correction method. The proposed method transfers only low-rank factors between the server and clients, hereby reducing the communications costs as compared to traditional methods. Furthermore, each client optimize over a small coefficient matrix. The memory, compute and communication costs of FeDLRT scale linearly with $n$.

The authors provide convergence result of their algorithm with full and simple variance correction schemes. To show the convergence guarantees to a stationary point, the work assume that the client and server-side loss functions are $L$-smooth but not necessarily convex. Finally, the authors demonstrate the efficiency of FeDLRT in an array of computer vision benchmarks and show a reduction of client compute and communications costs up to an order of magnitude with minimal impacts on global accuracy.

**Strengths:**

1) This work is well motivated. The challenges faced in horizontal federated learning are clearly stated and this work introduces a new algorithm to address these issues.

2) Computational results are promising.

3) Authors provide convergence results for their algorithm.

**Weaknesses:**

1) The authors compare the computational costs of their scheme with 3 algorithms in table 1. However, in their experiments their only compare with FedLin. It would be helpful to compare their work with the other algorithms.

2) The description of the algorithm (section 3.1) can be improved. The authors should try to explain each step of their algorithm better. Also Algorithm 2 (which is not an algorithm) is placed in the appendix. As far as I noticed, there is no mention of Algorithm 2 in main body of the paper. Yet Algorithm 2 contains the important definitions/notations that are used throughout section 3.1. The authors can consider adding all these concepts in Section 3.1 while they explain the algorithm.

**Questions:**

1) As mentioned in table 1, FeDLRT enjoys a lower computation, memory and communication costs that scales linearly with $n$ unlike other current algorithms. This is a significant strength of this work so it might be helpful to mention this in Section 1.

2) (line 210) If there is no page limit, it might be helpful to add the proof of lemma 1 or a sketch of the proof to the main body of the paper.

3) (line 210) If i understand correctly, using Lemma 1 the authors are able to reduce the communication costs. This could be explained in more detail.

4) (Figure 1) You can consider placing Figure 1 on page 6.

5) It would be interesting to see computational results with different singular value truncation threshold. This could be added or the effect could be explained briefly (if there is a page limit). Further, how is this selected?

6)  Many plots in this paper can be improved for neatness and clarity.

---

> ### Author Response · Authors · 2024-11-25
>
> We thank the reviewer for their thoughtful and constructive review. We appreciate that they found the work well
> motivated and the computational results with the proposed method promising. Please find below our answers to their
> feedback.
>
> ### W1 Comparison to other papers
>
> Among numerical results for heterogeneous data and a higher number of clients, we have compared FeDLRT (our method)
> against SCAFFOLD, FedGate, FedCOMGATE, and FePAQ in Appendix C.5 (and Page 8) of the revised manuscript, showcasing the
> competitive communication cost and
> optimization properties of our method. We remark that FeDLRT is able to reduce the client compute cost relative to
> FedAVG, whereas the competitor methods do not.
>
> ---
>
> ### W2 The description of the algorithm
>
> We remark that Algorithm 2 is referred to in the title of Algorithm 1 and contains the auxiliary functions
> for Algorithm 1. We decided on this format to increase readability of Algorithm 1 . We have included a remark at the
> first appearance
> of Algorithm 1 that points to the location of the auxiliary function definition.
>
> ---
>
> ### Q1 Linear Scaling
>
> Thank you for your positive assessment of our method; we have added a statement about the linear scaling in $n$ in the
> contribution paragraph of Section 1.
>
> ---
>
> ### Q2 Proof of Lemma 1
>
> Unfortunately, there is a page limit of 10 pages for an ICLR submission. To accommodate as much as possible of the
> feedback, we have to keep the proof of Lemma 1 in the appendix. This being said, the sketch of the proof of Lemma 1 is
> as follows.
> By construction, the augmented basis $\widetilde{U}$ is orthonormal and contains the old (orthonormal) basis $U$ in the
> first half of its columns. Since the second half, $\bar{U}$, of the augmented basis is orthogonal to the first half
> $U$ (the same argument holds for $V$), a projection of $S$ onto the augmented basis takes the block structure as seen in
> Lemma 1.
>
> ---
>
> ### Q3 Reduction of communication cost
>
> You are absolutely correct. Since we have a closed form for the augmented coefficient matrix
> $\widetilde{S}\in\mathbb{R}^{2r\times 2r}$ at the start of a local iteration, it is enough to communicate the original
> coefficient matrix ${S}\in\mathbb{R}^{2\times 2}$ and assemble the augmentation locally by padding with zeros, thus
> saving $75\%$ of the communication cost for this operation. Furthermore, the old basis $U$ has been sent to the client,
> so only the augmented part needs to be sent, reducing the communication cost by another $50\%$. This schematic is
> illustrated in Figure 2. We expanded the interpretation of Lemma 1 to improve readability.
>
> ---
>
> ### Q4 Figure placement
>
> The idea of the placement of Figure 1 is to directly illustrate the difference between variance corrected methods (
> FedLIN) and standard methods (FedAvG), when we present the literature review. To avoid confusion of the figure labels,
> we can accommodate the request to move the figure to the numerical result section at the end of rebuttal period if it is
> preferred.
>
> ---
>
> ### Q5 Singular Value Truncation
>
> Thank you for this remark. The truncation tolerance is a hyperparameter of dynamical low-rank schemes and essentially
> determines how aggressively one compresses the layers. Essentially, compressed neural network training creates a
> Pareto-front between compression rate and neural network accuracy. The hyperparameter $\tau$ determines the location of
> the model on this Pareto-front, where higher $\tau$ favors higher compression at the expense of accuracy and vice versa.
> $\tau$ determines the percentage of singular values (weighted by magnitude) that are truncated in each compression step.
> This truncation leads to an approximation error, which appears on the right hand side (the term with
> $\vartheta=\tau||\widetilde{S}^*||_F$) of the error bounds of Theorems 2, 3, 4 and Corollary 1. Since the focus of this
> work is the FL aspect, we have selected a standard value for $\tau$. An extensive study for the effect of $\tau$ on a
> dynamical low-rank training method is investigated in [1, 2].
>
> [1]Steffen Schotthöfer, Emanuele Zangrando, Jonas Kusch, Gianluca Ceruti, Francesco Tudisco; " Low-rank lottery tickets:
> finding efficient low-rank neural networks via matrix differential equations"
>
> [2]Emanuele Zangrando, Steffen Schotthöfer, Gianluca Ceruti, Jonas Kusch, Francesco Tudisco; " Geometry-aware training
> of factorized layers in tensor Tucker format"
>
> ---
>
> ### Q6 Plots
>
> We have moved Figure 6 (old manuscript) to the appendix (Figure 7 new manuscript) and increased the font size for
> readability. We are happy to implement any further suggestions to improve the illustrations of our manuscript.

---

> > ### Comment · Reviewer_P7tt · 2024-11-26
> >
> > Thank you for your response.

---

### Official Review · Reviewer_zAmN · 2024-11-04

**Soundness:** 2
**Presentation:** 2
**Contribution:** 2
**Rating:** 5
**Confidence:** 3

**Summary:**

This paper introduces a federated dynamical low-rank training (FeDLRT) approach aimed at reducing both compute and communication costs for clients in horizontal federated learning, addressing two key bottlenecks in this field. FeDLRT leverages dynamical low-rank splitting methods to construct a shared, global low-rank basis for network weights, enabling clients to train on a much smaller coefficient matrix rather than the full model. The global low-rank basis remains consistent across clients, allowing for an integrated variance correction scheme and ensuring convergence to a stationary point. Additionally, FeDLRT dynamically adjusts the rank through augmentation and truncation, optimizing resource usage in real-time.

The authors validate FeDLRT’s effectiveness on various computer vision benchmarks, demonstrating significant reductions—up to an order of magnitude—in both client compute and communication requirements, while maintaining high accuracy.

**Strengths:**

Overall, the paper is well-structured and clearly written, making it relatively easy to follow the development of ideas and contributions. The flow of the narrative allows readers to engage with the paper’s content without difficulty.

The inclusion of thorough discussions between technical sections, such as detailed explanations following each lemma and theorem, is particularly valuable. These discussions help clarify the purpose and implications of the theoretical results, making the technical content more accessible and aiding readers in understanding the significance of each result within the broader context of the paper.

The algorithm is presented effectively, with a well-organized description and clear pseudo-code. This clarity in presentation facilitates understanding of the method and makes it easier for others to replicate or build upon the approach.

Additionally, the paper includes an extensive experimental section in the appendix, showcasing a wide range of deep neural network (DNN) experiments. This supplementary section broadens the paper’s experimental validation, demonstrating the versatility and effectiveness of the proposed method across various architectures and datasets. The breadth of experiments contributes to a more comprehensive evaluation, which is especially beneficial for readers interested in practical applications of the theoretical findings.

**Weaknesses:**

1. **Abstract Length and Clarity**: The abstract is brief and could be expanded to provide a clearer, more comprehensive overview of the main ideas and contributions of the paper. Currently, it may be difficult for readers to fully grasp the paper’s objectives and significance from the abstract alone.

2. **Introduction and Literature Review**: Both the introduction and the literature review sections are quite short and lack depth. An expanded literature review would provide readers with a better understanding of the existing work in the field and how this paper’s contributions fit within that context.

3. **Client Drift Reduction Methods**: The paper does not discuss several important client drift reduction techniques, such as ProxSkip, S-Local-SVRG, and 5GCS. Notably, S-Local-SVRG predates FedLin and introduces essential concepts for handling client drift. The TAMUNA method also offers communication acceleration, variance reduction, and client drift correction through compression, which could provide useful context for this paper.
    - References:
        - Mishchenko, Konstantin, et al. "Proxskip: Yes! local gradient steps provably lead to communication acceleration! finally!." International Conference on Machine Learning. PMLR, 2022.
        - Hu, Zhengmian, and Heng Huang. "Tighter analysis for proxskip." International Conference on Machine Learning. PMLR, 2023.
        - Gorbunov, Eduard, Filip Hanzely, and Peter Richtárik. "Local sgd: Unified theory and new efficient methods." International Conference on Artificial Intelligence and Statistics. PMLR, 2021.
        - Grudzień, Michał, Grigory Malinovsky, and Peter Richtárik. "Can 5th generation local training methods support client sampling? yes!." International Conference on Artificial Intelligence and Statistics. PMLR, 2023.
        - Condat, Laurent, et al. "TAMUNA: Doubly Accelerated Federated Learning with Local Training, Compression, and Partial Participation." International Workshop on Federated Learning in the Age of Foundation Models in Conjunction with NeurIPS 2023.

Could you please add a comparison in the table that includes not only the mentioned papers but also other relevant works? This would provide a more comprehensive view of the existing literature and allow readers to better understand how your approach stands in relation to other methods.

4. **Compression in Federated Learning**: The literature review would also benefit from mentioning works on local methods for federated learning that incorporate compression. This line of research is relevant for addressing communication efficiency, an area where this paper aims to contribute.
    - Relevant papers:
        - Haddadpour, Farzin, et al. "Federated learning with compression: Unified analysis and sharp guarantees." International Conference on Artificial Intelligence and Statistics. PMLR, 2021.
        - Li, Zhize, et al. "SoteriaFL: A unified framework for private federated learning with communication compression." Advances in Neural Information Processing Systems 35 (2022): 4285-4300.
        - Meinhardt G. et al. Prune at the Clients, Not the Server: Accelerated Sparse Training in Federated Learning //arXiv preprint arXiv:2405.20623. – 2024.

5. **Notation and Formulation**: The notation used in the formulation is non-standard and may confuse readers. The symbol $ C $ is commonly used to denote the client cohort size (i.e., the mini-batch size of clients), while other symbols like $ M $ or $ n $ are often used for the total number of clients. Changing this notation would improve clarity.

6. **Client Drift Correction Communication Requirements**: The paper mentions that client drift correction requires additional communication. However, recent work, such as ProxSkip and SCAFFOLD, shows that client drift correction can be achieved without extra communication. Referencing these methods and discussing this point would clarify the communication requirements of different approaches.

    - Reference: Yu, Yaodong, et al. "TCT: Convexifying federated learning using bootstrapped neural tangent kernels." Advances in Neural Information Processing Systems 35 (2022): 30882-30897.

7. **Lipschitz Continuity of Gradients**: The paper's assumptions about Lipschitz continuity (or \( L \)-smoothness) of gradients are not rigorously defined. It would be helpful to clearly state these assumptions to ensure they align with the rest of the analysis.

8. **Clarity of Lemma 1**: The statement of Lemma 1 is somewhat unclear. To improve readability, it would be beneficial to make this lemma more rigorous and precise.

9. **Theorem 1**: Theorem 1 is also ambiguously stated, particularly regarding the meaning of the inequality provided. Adding a discussion to explain the significance of this inequality and how it relates to the paper's objectives would be beneficial.

10. **Theorem 2**: Similar to Theorem 1, Theorem 2 is stated in a way that may be confusing to readers. Providing a clear interpretation of the inequality and its implications would enhance understanding.

11. **Convergence Rate in Theorem 3**: In Theorem 3, the first term exhibits a convergence rate of \(1/T\), but the second term does not decrease (even with a diminishing step size). This contrasts with other methods that correct for client drift, which offer convergence guarantees to the exact minimizer without residual noise or error terms. It would be helpful to explain why this residual term persists in the analysis.

12. **Assumption 1**: Assumption 1 appears restrictive but lacks commentary. Including a discussion on the motivation and implications of this assumption would clarify its role in the analysis and inform readers about its necessity.

13. **Clarity of Theorem 4**: Theorem 4 is somewhat confusing, especially regarding the inequality it presents. A more thorough explanation of this inequality and its relevance to the paper’s objectives would be beneficial.

14. **Corollary 1 Convergence Rate**: Similar to Theorem 3, Corollary 1 includes a term with a convergence rate of \(1/T\) but also a second, non-diminishing term, even with a decreasing step size. Notably, other client drift-correction methods guarantee convergence to the exact minimizer without such persistent error terms. Discussing this difference would improve clarity.

15. **Connections to Strongly Convex or PL Cases**: The current theoretical results are limited to the general non-convex setting, which tends to yield conservative bounds. There are no results for strongly convex, convex, or PL-conditioned non-convex cases, though the first set of experiments is on a linear regression task, a convex problem. Clarifying how the theoretical findings connect to the experiments would enhance coherence between the theory and practical validation.

**Questions:**

Please check Weaknesses section

---

> ### Author Response · Authors · 2024-11-25
>
> We thank the reviewer for their thorough and constructive feedback. We appreciate that they found our work engaging and
> the techical contributions valueable.
> We have addressed the concerns ans questions below:
>
> ### W1 Abstract
>
> Thank you for the feedback. We have change the wording of the abstract to point out our contributions and their
> significance.
>
> ---
>
> ### W2 Introduction and Literature:
>
> Thank you for this remark. In the revised introduction section,
> we expanded the literature review and compared the proposed FeDLRT to existing FL methods. We also extended the last
> paragraph of section 2 and the beginning of section 3 to outline our contribution.
>
> ---
>
> ### W3 Client Drift Reduction Methods:
>
> We consider the primary contribution of FeDLRT is in the saving of compute, communication, and memory costs on both the
> server and client sides. The global basis introduced in FeDLRT enables straightforward incorporation of client drift
> reduction methods, e.g. FedLin. This feature differentiates FeDLRT from existing low-rank training methods, e.g., FedHM
> and FedPara. We discuss this in the last paragraph of Section 3 of the new manuscript.
>
> ---
>
> ### W4 Comparison to other approaches:
>
> Thank you for suggesting the comparison. We expand the introduction section to include a more detailed discussion about
> existing FL methods with low-rank and sparse compression and their comparison to FeDLRT.
>
> The primary difference between FeDLRT and many other low-rank or sparse compression methods, e.g., FeDLR, Comfetch, and
> FetchSGD, is that FeDLRT only trains the low rank coefficient matrix on the clients, which leads to much lower memory
> and compute costs on clients. On the other hand, by introducing the global basis, FeDLRT avoids the assembly of the full
> rank weight matrices on the server and leads to convergence guarantees, which are not available in existing methods that
> train only the low rank or sparse components (sub-models) on the clients, such as FedHM, FedPara, FjORD, and PriSM.
> The detailed comparison between FeDLRT and other methods in terms of compute, memory, and communication costs on both
> the server and client is now given in Table 1 in the manuscript.
> The TAMUNA method also offers communication acceleration, variance reduction, and client drift correction through
> compression. A key difference to the proposed FeDLRT method is that FeDLRT trains only the coefficient matrix on the
> clients, thus the training cost on the client is also reduced. The variance correction acts only on the client
> coefficient and is therefore efficient to compute as well. We have added this work in the related work section.
>
> ---
>
> ### W5 Notation:
>
> Thank you for this suggestion. We will change this notation in the revised manuscript in the end of the review period to
> avoid confusion about the notation **during** the reviewing period.
>
> ---
>
> ### W6 Client Drift Correction Communication Requirements
>
> Thank you for the suggestion. Indeed, incorporating other client drift correction schemes, such as ProxSkip and SCAFFOLD
> into FeDLRT could potentially eliminate the additional communication round required by the standard variance correction
> scheme in FedLin.
> This is one of the important future research direction that we plan to explore.
>
> ---
>
> ### W7 L-smoothness
>
> L- smoothness in this paper is defined as L-continuity of a function and its gradient, where both L constants are
> identical. That is for a function $f(x)$ with gradient $G(x)$, we have $|| f(x_1)-f(x_2) || \leq L || x_1 -x_2||$
> and $|| G(x_1)-G(x_2) || \leq L || x_1 -x_2||$ for some $L>0$. We are happy to incorporate this change in Appendix E.
>
> ---
>
> ### W8 Clarity Lemma 1
>
> Thank you for the suggestion. We have restated Lemma 1 in the revised manuscript.
>
> ---
>
> ### W9 Theorem 1
>
> Theorem 1 bounds the drift of the coefficient matrix $\widetilde{S}_c^s$ of client $c$ at local iteration $s$ by the
> global gradient given a learning rate restriction for $\lambda$. An explanatory statement is given in the text after
> Theorem 1. This bound allows us to state Theorem 2, a loss-descent guarantee for FeDLRT.
>
> ---
>
> ### W10 Theorem 2
>
> Theorem 2 show the global loss descent property of FeDLRT, up to the error $L\vartheta$ introduced by low rank
> truncation, where $\vartheta$ is the truncation tolerance and $L$ the Lipschitz constant of the loss function.
>
> ---

---

> > ### Author Response · Authors · 2024-11-25
> >
> > ### W11 Convergence Rate in Theorem 3
> >
> > Thank you for pointing this out. The error term in Theorem 13 is an artifact of the low-rank approximation rather than
> > the client drift correction. It comes from the additive term in $L\vartheta$ in Eq. (12).
> >
> > This truncation error is typical for low-rank training schemes in both the centralized (
> > e.g., https://arxiv.org/pdf/2205.13571 ) and FL (e.g., https://arxiv.org/pdf/2104.12416 ) settings.}
> >
> > ---
> >
> > ### W12 Assumption 1
> >
> > Assumption 1 implies that most of dynamics in the gradient flow are captured by the non-augmented basis. The assumption
> > holds when a suitable rank-$r$ manifold has been found in the training process, in which case the augmented basis does
> > not benefit the training.
> > To support the assumption, we record the gradients in the training process of ResNet18 on the CIFAR10 dataset using
> > FeDLRT and report the results in Figure 13 in the appendix of the revised manuscript.
> >
> > ---
> >
> > ### W13 Clarity of Theorem 4
> >
> > Theorem 4 is the adaptation of Theorem 2 for modified algorithm with simplified variance correction, which leads to a
> > slightly weaker loss descent guarantee for the simplified algorithm, as stated after Theorem 4 in the main manuscript
> >
> > ---
> >
> > ### W14Corollary 1 Convergence Rate
> >
> > Similar to Theorem 3, this non-diminishing term is the global low-rank approximation error. We remark that this error
> > exists in both the non-federated low-rank training setting (see the last term in Theorem 2
> > of {https://arxiv.org/pdf/2205.13571} ) and other low-rank FL methods (see the last term in Theorem 1
> > of https://arxiv.org/pdf/2104.12416 ). For detailed analysis of this low-rank truncation error, we refer
> > to [https://arxiv.org/pdf/2205.13571], Theorem 2.
> >
> > ---
> >
> > ### W15 Connections to Strongly Convex or PL Cases
> >
> > Thank you for this question. Although our analysis in this publication focuses on the non-convex case, which is most
> > relevant for the traget application of neural network training, convergence statements for convex, convex, or
> > PL-conditioned non-convex cases are possible. In particular, using the same techniques as in the proof of Theorem 2 and
> > 3, and following [https://arxiv.org/pdf/2102.07053] the results extend to the more structured cases.

---

> > > ### Comment · Reviewer_zAmN · 2024-11-27
> > > **Response**
> > >
> > > Thank you for addressing the issues and revising the draft. I appreciate the time and effort the authors have invested in improving the manuscript.
> > >
> > > However, I still believe there are some challenges regarding the clarity and the overall contribution of the paper. Nonetheless, I recognize the significant improvements made and have adjusted my score accordingly to reflect these efforts.

---

### Official Review · Reviewer_W6cB · 2024-11-04

**Soundness:** 2
**Presentation:** 3
**Contribution:** 2
**Rating:** 5
**Confidence:** 4

**Summary:**

The authors introduce a strategy (FeDRLT) for federated training which enables cheaper communication and client computational costs by training over low-rank decompositions of weights. Their algorithm includes a variance correction term to reduce client drift. The convergence of DRLT is $\mathcal{O}(1/T)$ with respect to minimal gradient norm. The authors perform LLSR and CIFAR-10 experiments of FeDRLT against FedLin and pure FedAvg, which performs respectably.

**Strengths:**

-Very clear narrative and motivation. Bi-directional compression is very desirable for FL and still a relatively open problem.

-The experiments perform very well against full-sized counterparts. The variance correction alleviates client drift and enables greater numbers of client participation.

-The convergence rate is good and clear proofs are provided for all guarantees.

-The technique seems like a new approach to low-rank FL and comes equipped with an interesting variance reduction mechanism. As bi-directional compression is still relatively rare in the compressive FL landscape, this work could be valuable, but please see weakness.

**Weaknesses:**

-The work is at worst incremental and at best, not adequately compared against similar efforts. The authors must compare against one or more compression works which shrink the gradient and/or weights. These works include FetchSGD [1] (gradient compression), ComFetch [2] (bi-directional compression + correction), Fjord [3] (model splitting), LASER [4] (gradient compression, HeteroFL [5] (model splitting), PriSM [6] (model splitting), etc. The field of compressive FL is rich (and saturated) and the methods described in the paper are reminiscent of other approaches.

-The variance correction mechanism is not a novel approach for compressed federated learning. In FL, sometimes this approach is referred to as "error feedback," and is necessary for most compression techniques [5] to compensate for reconstruction error, but doubles as a client-drifting mechanism as it is aggregated over all clients. See [6] for another mechanism and general discussion of variance reduction for distributed SGD. The authors should comment their approach in light of this existing literature.

-The experiments are weak. LLSR is insufficient nowadays to demonstrate the efficacy of an FL algorithm. Although CIFAR-10 and CIFAR-100 experiments are conducted, it does not appear that there is any ablation over varying degrees of heterogeneity, which is absolutely necessary for a work which is attempting to combat client drift. The authors should ablate over varying Dirichlet allocations of classes and compare their algorithm to a few other compression works or at least mimic their setups and compare cited numbers.

[1] Rothchild, D., Panda, A., Ullah, E., Ivkin, N., Stoica, I., Braverman, V., ... & Arora, R. (2020, November). Fetchsgd: Communication-efficient federated learning with sketching. In International Conference on Machine Learning (pp. 8253-8265). PMLR.

[2] Rabbani, T., Feng, B., Bornstein, M., Sang, K. R., Yang, Y., Rajkumar, A., ... & Huang, F. (2021). Comfetch: Federated learning of large networks on constrained clients via sketching. arXiv preprint arXiv:2109.08346.

[3] Horvath, S., Laskaridis, S., Almeida, M., Leontiadis, I., Venieris, S., & Lane, N. (2021). Fjord: Fair and accurate federated learning under heterogeneous targets with ordered dropout. Advances in Neural Information Processing Systems, 34, 12876-12889.

[4] Makkuva, A. V., Bondaschi, M., Vogels, T., Jaggi, M., Kim, H., & Gastpar, M. C. (2023). Laser: Linear compression in wireless distributed optimization. arXiv preprint arXiv:2310.13033.

[5] Ivkin, N., Rothchild, D., Ullah, E., Stoica, I., & Arora, R. (2019). Communication-efficient distributed SGD with sketching. Advances in Neural Information Processing Systems, 32.
[6] Liang, X., Shen, S., Liu, J., Pan, Z., Chen, E., & Cheng, Y. (2019). Variance reduced local sgd with lower communication complexity. arXiv preprint arXiv:1912.12844.

**Questions:**

See Weaknesses

---

> ### Author Response · Authors · 2024-11-25
>
> Thank you for the constructive review. We appreciate that you find our narrative and motivation helpful and found our
> experiments insightful. Please find our responses to your concerns below:
>
> ### W1. **Comparison to other approaches**:
>
>
> Thank you for suggesting the comparison. We expand the introduction section to
> include a more detailed discussion about existing FL methods with low-rank and sparse compression and their comparison
> to FeDLRT.
>
> The primary difference between FeDLRT and many other low-rank or sparse compression methods, e.g., FeDLR, Comfetch, and
> FetchSGD, is that FeDLRT only trains the low rank coefficient matrix on the clients, which leads to much lower memory
> and compute costs on clients. On the other hand, by introducing the global basis, FeDLRT avoids the assembly of the full
> rank weight matrices on the server and leads to convergence guarantees, which are not available in existing methods that
> train only the low rank or sparse components (sub-models) on the clients, such as FedHM, FedPara, FjORD, and PriSM.
> The detailed comparison between FeDLRT and other methods in terms of compute, memory, and communication costs on both
> the server and client is now given in Table~1 in the manuscript.
>
> ---
>
> ### W2. **variance correction**:
>
>
> We agree that the variance correction or the error feedback mechanism is not novel.
> One of the contributions of FeDLRT is that, by introducing the global low rank basis, FeDLRT enables the incorporation
> of variance correction mechanisms. To the best of our knowledge, other FL schemes training only on low rank factors,
> e.g., FedHM and FedPara, are unable to use variance correction effectively.
>
> We included the suggested variance correction methods in the discussion of variance correction techniques in Section 2
> together with FedLin, which we focus on.
> We also commented in the future work section on the potential extension of FeDLRT to accommodate variance correction
> approaches other than the one used in FedLin. We expect that, with the global basis in FeDLRT, it would be
> straightforward to incorporate other variance correction scheme and potentially improve the computation and
> communication efficiency of FeDLRT.
>
> ---
>
> ### W3. **heterogeneous data distribution***
>
>
> As requested, we provide experiments with ResNet18 on CIFAR-10 in a heterogeneous case using Dirichlet split with
> parameter $\alpha\in\{10,5,2,1\}$. The results are ported in Figure~6 of the revised manuscript. We compare FeDLRT with
> variance correction to FedLin using the training hyperparameters described in Table 2 in Appendix D. We choose $C=16$
> clients with $10$ local iterations and train for $2000$ aggregation rounds. The numerical results show that, as in the
> data homogeneous case, FeDLRT mirrors the performance of FedLin while reducing the compute, communication, and memory
> costs.

---

### Official Review · Reviewer_3Wm9 · 2024-11-04

**Soundness:** 3
**Presentation:** 2
**Contribution:** 2
**Rating:** 5
**Confidence:** 3

**Summary:**

The paper introduces the FeDLRT algorithm to jointly reduce client compute and communication cost in Federated Learning (FL). The method consist in estimating a global low-rank basis of network weights and constraing local client optimization to that manifold, with an optional variance-reduction term applied to local updates.
The work presents a convergence proof of the proposed algorithms and variants for non-convex objectives in full participation, demonstrating that FeDLRT achieves state-of-art linear convergence rate.
Experiments include theoretical settings (linear least squares regression) and computer vision setup (CIFAR-10 with ResNet-18/AlexNet/VGG and CIFAR-100 with ViT), and are convincing in showing the effectiveness of the proposed approach.

**Strengths:**

- The approach has strong theoretical guarantees, and the empirical results seems matching the theory;
- The method jointly reduces communication cost and client compute, since clients only optimize smaller matrices w.r.t. full weights

**Weaknesses:**

- **Poor introduction of the premises for the algorithm:** in my opinion the second part of section 2 is not clear enough in explaining the concepts the proposed algorithm is based on. In particular, it is not evident which technical innovations this work brings with respect to previous works. Authors are encouraged to dedicate more space to discuss how this work build on previous concepts and how it advances the state of the art.
- **Poor comparison with other approaches:** the proposed method is compared essentially only with FedLin and FedAvg, while many methods exist in the categories the authors outline in the introduction ((i) methods that only communicate compressed updates and (ii) methods that reduce by also learning low-rank factors). Other related methods in this area are correctly cited but not well-discussed and not considered in the evaluation, such as FedPara [1], FedDLR [2], FEDHM [3]. Additionally, related papers that I believe should be included and discussed are LBGM [4] and FetchSGD [5].
Another line of works that should be considered is subnetwork training in FL, for example FjORD [6], which achieves training and communication efficiency by locally training subnetwork via Ordered Dropout.



[1] FedPara: Low-rank Hadamard Product for Communication-Efficient Federated Learning, ICLR 2022

[2] Communication-Efficient Federated Learning with Dual-Side Low-Rank Compression

[3] FedHM: Efficient Federated Learning for Heterogeneous Models via Low-rank Factorization

[4] RECYCLING MODEL UPDATES IN FEDERATED LEARNING: ARE GRADIENT SUBSPACES LOW-RANK?, ICLR 2022

[5] FetchSGD: Communication-Efficient Federated Learning with Sketching, ICML 2020

[6] FjORD: Fair and Accurate Federated Learning under heterogeneous targets with Ordered Dropout, NeurIPS 2021

**Questions:**

- Could you better describe the relationship with respect to the above related works both regarding the methodology and the convergence guarantees?
- What are the assumptions regarding statistichal heterogeneity for the convergence of FeDLRT? I did not find an equivalent of the common "bounded gradient dissimilarity" assumption for the low-rank training of the proposed method, and I wonder what are the guarantees in heterogeneous FL
- With respect to the previous question, can you propose some experiments similar to the ones presented for real-world dataset (e.g. CIFAR-10 with ResNet18) in heterogeneous cases (e.g. Dirichlet split as standard)? I would like to see an assessment of the performance of the proposed methods in non-convex case on real-world dataset, as opposed to the (valuable) results already provided for the linear squares regression in figure 1.

**Other suggestions:**
- **Presentation of theoretical results:** Theorems 1,2 and 5 should be lemmas in my opinion, since they are intermediate results necessary for the convergence rates. Please evaluate providing also a more in-depht discussion of these results
- **Organization of figures:** figure 6 in hardly readable in a printed version of the manuscript. Please evaluate enlarging the font size as well as the size of the overall figure. Alternatively, results in tabular form could be an option. I also suggest not to put wrap figure 4 in the text, as it makes the text less readable.
- **Empirical validation of assumption 1:** I think that and empirical validation of assumption 1 could benefit the clarity and the robustness od the paper. While the assumption seems reasonable in homogeneous cases and the corresponding methods seems pretty equivalent in performance in the provided experiments, I expect statistical heterogeneity to play a role in this matter. Could you plot the quantity in the LHS of eq. 14 in one of the real-world experiments of your setup (e.g. CIFAR-10 with ResNet18), in iid and non-iid conditions?

---

> ### Author Response · Authors · 2024-11-25
>
> Thank you for the constructive review. We appreciate you find our approach convincing with strong theoretical guarantees
> and matching empirical results. Please find below the itemized answers to your questions.
>
> ---
>
> ### W1. **Technical innovation and comparison to existing work**:
>
> Thank you for pointing out the clarity issue in Section 2
> of our manuscript. We have added a paragraph at the end of Section 2 to address this. The changes in the manuscript are
> marked blue. The idea of this addition is summarized below.
>
> The main technical innovation of our manuscript is to lift the Dynamical Low-Rank Training (DLRT) framework
> of  [Schotthöfer et al. (2022); Zangrando et al. (2023); Hnatiuk et al. (2024)] to the federated learning scenario.
>
> **Direct extension of DLRT to a federated setup is not straightforward.**
> In a FL setup, applying the DLRT scheme to the local training problem on
> each client $c$ leads to low-rank weights
> $W_{c} = U_cS_cV_c^\top$ with different bases $U_c$ and $V_c$ and potentially different ranks for each client. While these
> factors can still be efficiently communicated, aggregating these low-rank weights on the server requires reconstructing
> the full weight matrix
> $W^*=\frac{1}{C}\sum_{c=1}^C U_cS_cV_c^\top$ . In this process, the low rank structure is lost and
> needs to be costly recovered by a full $n\times n$ SVD on the server.
>
> **The proposed FeDLRT method addresses this issue by constructing global basis $U$ and $V$ that are shared among
> all clients.**
>
> FeDLRT builds a global orthogonal basis $U,V$ and only evolves the coefficient matrix $\tilde{S_c}$
> on each client
> $c$.
> Therefore the aggregation step changes from $W^*=\frac{1}{C}\sum_{c=1}^C U_cS_cV_c^\top$ to Eq. (10) in the manuscript,
> i.e.  $\tilde{W_r^*}=\tilde{U}(\frac{1}{C}\sum_{c=1}^C \tilde{S}_c)\tilde{V}^\top.$
> Here the rank of $\tilde{W}_r^*$ is at most $2r$ and only requires a $2r\times
> 2r$ SVD on the server to update the basis, which is much cheaper than the $n\times n$ SVD needed for the full weight
> matrix $W^*$. Furthermore, evolving only the coefficient matrix $\tilde{S}_c$ on each client $c$ significantly
> reduces the compute costs on each client.
>
> The global basis also allows us to establish loss descent and convergence properties of FeDLRT, maintain a low rank
> during aggregation, and leverage variance correction methods.
> In particular, without the global basis, the application of standard variance correction scheme is not straightforward.
>
> ---
>
> ### W2. and Q1. **Comparison to other approaches**:
>
> Thank you for suggesting the comparison. We expand the introduction
> section to include a more detailed discussion about existing FL methods with low-rank and sparse compression and their
> comparison to FeDLRT.
>
> The primary difference between FeDLRT and many other low-rank or sparse compression methods, e.g., FeDLR, Comfetch, and
> FetchSGD, is that FeDLRT only trains the low rank coefficient matrix on the clients, which leads to much lower memory
> and compute costs on clients. On the other hand, the global basis introduced in FeDLRT avoids the assembly of the full
> rank weight matrices on the server and leads to convergence guarantees, which are not available in existing methods that
> train only the low rank or sparse components on the clients, such as FedHM, FedPara, and FjORD.
> The detailed comparison between FeDLRT and other methods in terms of compute, memory, and communication costs is now
> given in Table~1 in the manuscript.
>
> ---
>
> ### Q2. **Assumptions on data statistics**:
>
> The current convergence analysis of FeDLRT does not require specific assumptions on the data distribution. Indeed, the
> variance correction term is introduced to mitigate statistical heterogeneity of the client data. We expect that, without
> variance correction, assumptions on gradient dissimilarity are likely needed to guarantee the convergence of FeDLRT.
> However, we only consider FeDLRT with variance correction in the current convergence analysis.
>
> ---
> ### Q3. **Experiments in data heterogeneous cases**
>
> As requested, we provide experiments with ResNet18 on CIFAR-10 in a heterogeneous case using Dirichlet split with
> parameter $\alpha= 10,5,2,1 $. The results are ported in Figure 6 of the revised manuscript. We compare FeDLRT with
> variance correction to FedLin using the training hyperparameters described in Table 2 in Appendix D. We choose $C=16$
> clients with $10$ local iterations and train for $2000$ aggregation rounds. The numerical results show that, as in the
> data homogeneous case, FeDLRT mirrors the performance of FedLin while reducing the compute, communication, and memory
> costs.

---

> > ### Author Response · Authors · 2024-11-25
> >
> > ### O1. **Presentation of theoretical results:**
> >
> > We agree that Theorem 1 can be changed to a lemma. However, in our opinion, the monotonicity property (up to truncation
> > error) given in Theorem 2 justifies it as a standalone theorem.
> > This property distinguishes FeDLRT from other FL schemes that train on the low-rank factors, e.g., FedHM and FedPara,
> > and is a result of the global basis featured in FeDLRT.
> > We added a statement to the main manuscript, that highlights this remark.
> > We will change Theorem~1 to a lemma in the very end of the rebuttal period, to not stir confusion about the changed
> > numbering of theorems in the main manuscript.
> >
> > Theorem 5 (of Appendix, Page 10) is one of the main theoretical results in the paper we cite it from. It shows that, in
> > DLRT, low rank training could lead to similar weights to the ones from regular full rank training, which gives models of
> > similar accuracy. In the context of FeDLRT, it implies that FeDLRT (FeDLRT with variance correction) mirrors the
> > trajectory of FedAvg (FedLin). Thus, similar validation accuracy values can be expected, as verified in the experiments.
> >
> > ---
> >
> > ### O2. **Organization of Figures**:
> >
> > To account for the page limit, we have put Figure 6 (old manuscript) in the
> > appendix (Figure 7 new manuscript) and increased the font size to increase readability.
> >
> > We experimented with a tabular presentation of Figure 6 (old manuscript). However, in the current representation as
> > figure, it is easier to see that the FeDLRT mirrors the validation accuracy performance of the full rank FedAvg and
> > FedLin schemes in one glance, thus we would like to keep it as a Figure.
> >
> > We will further move Figure 4 to the appendix to improve readability and use the gained space to accommodate the
> > explanations requested by the reviewers.
> >
> > ---
> >
> > ### O3. **Empirical validation of Assumption 1.**
> >
> > We illustrated the empirical evaluation of Assumption 1 for ResNet18
> > on CIFAR10 in Figure 13 in the appendix of the revised manuscript.

---

> ### Comment · Reviewer_3Wm9 · 2024-12-01
>
> I would like to thank the authors for their rebuttal and for having updated the manuscript. On my side there are still critical concern I would like the authors to address.
>
> **W1: Comparison with previous works**
>
> > Direct extension of DLRT to a federated setup is not straightforward.
> I acknowledge the difference w.r.t. DLRT, and I believe it is a nice property to have. However, I cannot fully justify the computation perspective as motivation on its own for proposing the method. While it is true that an additional SVD is necessary if trivially extending DLRT to FL, it is also true that is would be computed on the server, which we can reasonably assume to be computationally powerful enough to handle the operation.
>
> > On the other hand, the global basis introduced in FeDLRT avoids the assembly of the full rank weight matrices on the server and leads to convergence guarantees, which are not available in existing methods that train only the low rank or sparse components on the clients.
>
> I read the additional paragraph at the end of section 2, but I am still not conviced by which is claimed above. In particular I did not find a comparison with LBGM [4], which similarly send only the singular values of a global basis of network weights, and does provide convergence guarantees. Please discuss in detail the relationship with [4].
>
> **Q2: Assumptions for convergence guarantees**
>
> > We expect that, without variance correction, assumptions on gradient dissimilarity are likely needed to guarantee the convergence of FeDLRT.
>
> This statement is too imprecise to be considered as valid, if the algorithm is able to obtain convergence under unbound heterogeneity (i.e. bounded gradient dissimilarity assumption is not used in the convergence proof) it should be clear where this comes from. Do the results of the analysis hold also under partial participation or full participation is assumed?Could you please elaborate on these points?
>
> **Q3: Experiments**
> Thanks for the additional experiments, they address my original question.

---

> > ### Author Response · Authors · 2024-12-01
> >
> > Thank you for the response and the further comments. We address them below.
> >
> > **W1** :
> >
> > - Comparing to the direct extension of DLRT, avoiding the full SVD computation at the server is not the only benefit of
> >   the global basis introduced in FeDLRT. Other advantages of introducing the global basis include
> >     - a) reduced memory
> >       requirement on the server (from $\mathcal{O}(n^2)$ to $\mathcal{O}(2nr)$), since the full weight matrices are
> >       never
> >       assembled,
> >     - b) reduced client-to-server communication cost (no need to upload bases to server),
> >     - c) reduced client
> >       computation cost (no need to update bases during local training), and
> >     - d) enabling client drift mitigation
> >       techniques.
> >
> >   We believe these benefits warrant the development of FeDLRT. We will make these points more clear in the camera ready
> >   version of the manuscript if the paper is accepted.
> >
> >
> > - It seems to us that the main idea of the LBGM method proposed in [4] is to compress the communication from clients to
> >   the server by reusing existing local gradient directions on the server. Specifically, when the new local gradient
> >   direction is close to the existing one, LBGM communicates from the client to the server only the (scalar) magnitude of
> >   the new local gradient projected to the existing direction, as opposed to communicating to the full local gradient.
> >   Comparing the LBGM, FeDLRT has the following advantages:
> >
> >     - a) reduced server and client memory cost, since the
> >       parameters (weights) are not compressed in LBGM,
> >     - b) reduced server-to-client communication (broadcasting) cost,
> >       because LBGM requires broadcasting full weight matrices after each aggregation,
> >     - c) lower client computation cost (
> >       FeDLRT trains the low rank coefficients only, while LBGM trains the full weights).
> >
> >   Further, the convergence analysis of LBGM assumes bounded dissimilarity of local loss functions whereas the
> >   convergence property of FeDLRT relies on the variance correction technique (see response to Q2 below for more
> >   details).
> >   We will add LBGM [4] into the method comparison in the camera ready version of the manuscript if the paper is
> >   accepted.
> >
> > ---
> >
> > **Q2**:
> >
> > The convergence of FeDLRT presented in this paper does not require the assumption of bounded dissimilarity of local loss
> > functions because, in the FeDLRT analysis, the dissimilarity of local loss functions is taken cared of by **the
> > variance correction scheme** , which was initially designed in FedLin to mitigate client drift.
> > If FeDLRT is performed _without_ variance correction, we expect that the assumption of bounded dissimilarity of
> > local loss functions may become necessary for convergence.
> > However, without variance correction, we observe suboptimal performance of FeDLRT when the data distribution is
> > heterogeneous (the model accuracy deteriorates when the loss function dissimilarity increases). Therefore, we only
> > analyze the case when FeDLRT is equipped with proper variance correction techniques to prevent such performance
> > degradation.
> >
> > The convergence analysis in this paper considers the full participation scenario. Given the encouraging numerical test
> > results on the partial participation case (see Figure 5 in the revised manuscript), we plan to extend the analysis to
> > the partial participation case, along with the incorporation of other potentially more efficient variance correction
> > schemes.

---

### Author Response · Authors · 2024-12-02

Dear Reviewers and Area Chair,

We thank the reviewers for their constructive feedback and insightful discussions.
As the discussion period concludes, we would like to summarize the key updates made to the manuscript in response to feedback:
1. Extended literature review:
    * We have expanded the introduction to better highlight our contribution and the features that distinguish FeDLRT from existing methods.
    * Table 1 has been updated to include a more comprehensive comparison of the compute, communication, and memory costs with related work.
2. Clarifications to the proposed method:
    * We have clarified the differences between the proposed FeDLRT and existing methods at the end of Section 2.
    * Additional statements in Section 3 have been included to enhance the explanation of the proposed methodology.
3. New test cases:
    * Partial client participation: Figure 5 reports the training result of FeDLRT with 10 out of 200 clients participating in each aggregation round. The results demonstrate that FeDLRT is applicable to partial participation scenarios and achieves accuracy comparable to non-compressed methods, e.g. FedAvg.
    * Heterogeneous data: Figure 6 shows that FeDLRT handles heterogeneous data (under Dirichlet distribution) effectively, achieving higher accuracy than FedLin with up to 62% compression rates.
    * Comparison with existing methods: Table 7 compares FeDLRT to state-of-the-art methods in terms of global training loss and communication cost. FeDLRT achieves comparable performance to FedCOMGATE (Haddadpour et al., 2020a) while reducing client compute costs.


We believe that these revisions, along with the additional results, have addressed the comments and strengthened the manuscript. We kindly hope this will lead to greater confidence and an improved rating. Should you have any remaining questions or require further clarifications, please do not hesitate to reach out.

---

### Meta-Review · Area_Chair_Kzdk · 2024-12-18

**Metareview:**

This work introduces FeDLRT, a federated dynamical low-rank training scheme that reduces client compute and communication costs in horizontal federated learning. By leveraging manifold-constrained optimization, FeDLRT trains a small coefficient matrix per client using a global low-rank basis, with dynamic augmentation and truncation to optimize resource utilization. The method ensures global loss descent and convergence, while maintaining accuracy comparable to traditional methods like FedAvg and FedLin on computer vision benchmarks.

Most reviewers indicate that paper introduces an interesting method for low-rank training in the federated setting. However, a concern shared by several reviewer is the lack of clarity in in comparing the proposed method with other state-of-art algorithms.

The AC shares these concerns and believes the paper would benefit from a major revision to address the highlighted issues.

**Additional Comments On Reviewer Discussion:**

The AC engaged in a thorough discussion with the reviewers and concluded that, while the paper has valuable contributions, it requires substantial revision beyond what is feasible within the submission timeline. The AC strongly encourages resubmission, recognizing the paper's importance and potential impact. This was truly a "borderline" decision, and the AC appreciates the authors' efforts and contributions, and hopes they remain encouraged, as the paper has strong merits and can make a significant contribution to the community once revised.

---

### Decision · Program_Chairs · 2025-01-22

Reject